# The effect of seasonal temperatures on the physiology of the overwintered honey bee

Olga Frunze[1], Yumi Yun[1], Hyunjee Kim[1], Ravil R. Garafutdinov[2], Young-Eun Na[3]*, Hyung-Wook Kwon[1]*

1 Department of Life Sciences & Convergence Research Center for Insect Vectors (CRCIV), Incheon National University R&D Complex, Yeonsu-gu, Incheon, Republic of Korea, 2 Institute of Biochemistry and Genetics, Ufa Federal Research Center, Russian Academy of Sciences, Ufa, Bashkortostan, Russian Federation, 3 Agro-materials Industry Division, Rural Development Administration (RDA), Wanju, Republic of Korea

☯ These authors contributed equally to this work.
* hwkwon@inu.ac.kr (HWK); youngman99@korea.kr (YEN)

**Data Availability Statement:** All relevant data are within the manuscript and its Supporting information files.

## Abstract

Honey bee physiology follows an annual cycle, with winter bees living ten times longer than summer bees. Their transition can be disrupted by climate change. Several climate factors, mainly temperature, may contribute to the global losses of winter bees. We simulated global warming by maintaining constant temperatures of 25°C (Group 25) and 35°C (Group 35) in rooms around hives from June to October, while a Group control experienced natural conditions. Colony performance was assessed in August and September. In February, workers were examined for physiological traits (acinus size and lipid content in the fat body) and molecular markers (*vg* and *JHAMT*), along with potential markers (*ilp1, ilp2, TOR1,* and *HSP70*). Our findings suggest that temperature decreases around winter worker broods from Group 25 in the fall led to their different physiological states related to aging in winter compared to Group 35 workers. Changes in bees from Group 35 the end of diapause were detected with an upregulation of *HSP70, ilp2,* and *TOR1* genes. These signs of winter bees in response to summer global warming could lead to the development of strategies to prevent bee losses and improve the identification of physiological states in insect models.

## Introduction

Honey bees (below bees) play a crucial role in human life as pollinators on agricultural farms, producers of honey, royal jelly, and other products, which are valued for their health and medicinal benefits, and contributors to ecosystems worldwide. However, the loss of bees has been reported over the last two decades, particularly at the end of the overwintering period [1–3].

Several factors arising during the year can contribute to this [4]. For example, climate warming or cooling [5] impacts the lifespan in flies [6], fish [7], bees [8], and rodents [9]. Other factors related to climate change include the quality of forage [10, 11], beekeeper management practices [12, 13], and pesticide exposure from land use [11]. Although we can try to

**Funding:** Yes Hyung Wook Kwon received. This work was carried out with the support of the Cooperative Research Program for Agriculture Science & Technology Development (Project No. RS-2023-00232749) and the Priority Research Centers Program through the National Research Foundation of Korea (NRF) funded by the Ministry of Education (2020R1A6A1A03041954). The funders had no role in study design, data collection and analysis, decision to publish, or preparation of the manuscript.

**Competing interests:** The authors have declared that no competing interests exist.

**Abbreviations:** AHC, Agglomerative Hierarchical Clustering; DA, discriminant analysis; FD, Fisher distance; HG, hypopharyngeal glands; HSP70, heat shock protein 70; ilp1, insulin-like peptide 1; ilp2, insulin-like peptide 2; JH, juvenile hormone; JHAMT, Juvenile Hormone Acid Methyltransferase; RpL32, Ribosomal protein L32; TOR1, target of rapamycin 1; vg, vitellogenin.

avoid some of these factors, no effective methods to mitigate the impacts of climate change and prevent bee winter mortality have been reported.

To advance in this direction, it is necessary to clarify the general processes occurring in late summer and fall that are involved in the transition from short-lived to long-lived workers [14, 15], as these processes may be disrupted by climate change. This transition has been associated with external (environmental) factors that influence internal (metabolic) processes.

Known external trigger factors include a decrease in pollen availability [16, 17] and cessation of brood rearing [15, 18]. Ambient temperature, a main factor of climate change, can significantly alter brood temperature in summer [19], despite bees'efforts to maintain a consistent brood microclimate [20–22].

With the onset of fall, external trigger factors initiate the transition of bees through internal processes, leading to the development of winter workers with prolonged longevity, which can last up to 8 months [1]. Briefly, this transition exhibits the classic insect diapause phenotype, characterized by reduced metabolic activity and altered hormonal profiles [14, 23–26]. Unlike other insects, the reduced metabolic activity of bees during diapause [17] includes contracting their wing muscles to generate heat and maintain warmth within the winter cluster. This process depends on a constant energy source and requires a pure carbohydrate diet [27], which is not necessary for other diapausing insects.

In winter bees, the physiology of internal processes related to this hormone-regulated phenomenon is well-described. It manifests as hypertrophied secretory vesicles and increased acinus diameter in the hypopharyngeal gland (HG) compared to newly emerged and forager summer bees [28]. HG development is known to be positively correlated with vitellogenin (*vg*) throughout a worker's lifespan. However, *vg* is synthesized in the fat body [29], which is more developed in winter bees compared to summer bees [27]. The fat body mass results from the accumulation of nutrients [30], with lipids constituting >90% of the mass in the form of triglycerides. These triglycerides are synthesized from dietary carbohydrates and fatty acids and are used not only for *vg* synthesis [30] but also for producing pheromones, phospholipids, and wax [31].

Significant differences between bees and other insects have been observed in the mutually inhibitory interaction between vg (protein) and the juvenile hormone (JH) [32]. An increased level of vg (protein and gene), alongside decreased levels of the JH [1, 15, 33], has been noted, however, the molecular mechanism of this complex relationship remains unclear [34]. It is known that the JH biosynthetic pathway begins in the glands in the brain complex and responds to sensory inputs that underpin the division of labor [34, 35]. Furthermore, at the end of winter, diapausing bees gradually transition to nursing brood care and foraging physiology [1]. The last transition is characterized by several physiological and molecular changes, including a decrease in acinus size [36] and fat body mass [37], as well as reductions in vg levels and increases in JH levels [15, 38]. These changes contribute to seasonal effects and behavior [39], leading to reduced longevity and increased aging compared to diapause conditions [40, 41].

Alongside with the well-known age-related marker gene *vg* [32, 42–44], this study examined other potential markers, including the genes *JHAMT*, *ilp1*, *ilp2*, *TOR1*, and *HSP70*. The enzyme involved in JH biosynthesis, *Juvenile Hormone Acid Methyltransferase (JHAMT)*, exhibits changes in expression similar to JH titer and can serve as an alternative marker to JH [45–47]. A positive correlation with JH was observed in the *insulin-like peptide*, *ilp1*, whose expression increases in the brains of foragers and decreases with continued foraging [48]. The *insulin-like peptide*, *ilp2*, gene may have task-related expression as it is highly expressed in nurse tissue and plays a role in increasing cell size and number in individual organs [49]. Additionally, the *TOR1* gene, a nutrient-sensing kinase, along with the *ilp1* and *ilp2* genes,

participates in the mobilization of nutritional resources (autophagy) from tissues such as the fat body and glands in *Drosophila* [50]. Moreover, the *TOR* pathway is implicated in disease and nutrition in insects [51–53] and mammals [54], and it also regulates longevity extension in social insects [25]. Lastly, the stress marker *heat shock protein 70 (HSP70)* was chosen because it is detected at low levels in nurses but shows increased expression in foragers [55]. While all these markers are expected to be important for identifying the physiological state in overwintered honey bees, testing them together in the same experiment requires evaluating their responses to the same triggers.

This research aimed to simulate the influence of global warming on colony performance in previous seasons and to compare the physiological state of overwintered *Apis mellifera ligustica* bees. The experiment involved maintaining colonies at ambient constant temperatures of 25°C (Group 25) and 35°C (Group 35) in controlled rooms, representing the average and high summer temperatures in Incheon (Republic of Korea), respectively. A Group control was kept under natural conditions. The temperature experiments continued throughout the summer, September, and October, with the colonies overwintering under the same conditions. Colony performance, including the brood microclimate, was assessed in August and September, during the transition from summer to winter physiology in the Republic of Korea. In February, we compared the long-term effects of ambient temperature on the physiological and molecular state of the winter bees at the end of diapause (broodless period) using eight markers. The two physiological markers were acinus size and lipid content in the fat body. The six molecular markers were for: tasks (*vg* and *JHAMT*), nutrition (*ilp1*, *ilp2*, and *TOR1*), and stress response (*HSP70*). We believe this study can lead to further research to identify the environmental causes and prevent honey bees losses early.

## Methods

### Experimental bees

Western bees (*Apis mellifera ligustica*) were obtained from nine colonies kept at the Incheon National University apiary in the Republic of Korea (Figs 1–4), managed by a professional beekeeper following standard beekeeping practices [56]. The experimental colonies contained first-year young sister queens, naturally mated, which started to lay eggs in May 2022. Prior to the experiment in June, colonies were standardized in terms of the number of frames (total 4), brood population (3 frames), honey amount (1 frame), and the number of bees (around 3000~3500 bees). Colony strength was assessed through visual inspection of adult population and brood area both before and during the experiment [56]. Two rooms equipped with a climate-control system were established to maintain bee colonies at 25°C (Group 25) and 35°C (Group 35), simulating global warming without day and night temperature differences.

These temperatures were selected to mimic the effects of global warming, with 25°C representing the summer average and 35°C being the highest recorded in Incheon city under typical conditions. However, the highest temperature was not naturally reached during the summer of 2022 (Fig 5). Other environmental conditions were not controlled. The ceiling light was used during colony inspections. In the third room housing bee colonies, the control group was provided with free ventilation without additional heating. All bees had equal opportunities to collect pollen and nectar around the apiary.

On September 18th and 25th, 2022, two combs with sealed brood from each colony were placed in an incubator set at 34°C. Upon hatching, around 200 workers were individually marked with colored paint on their thoraxes and then reintroduced into their original colony.

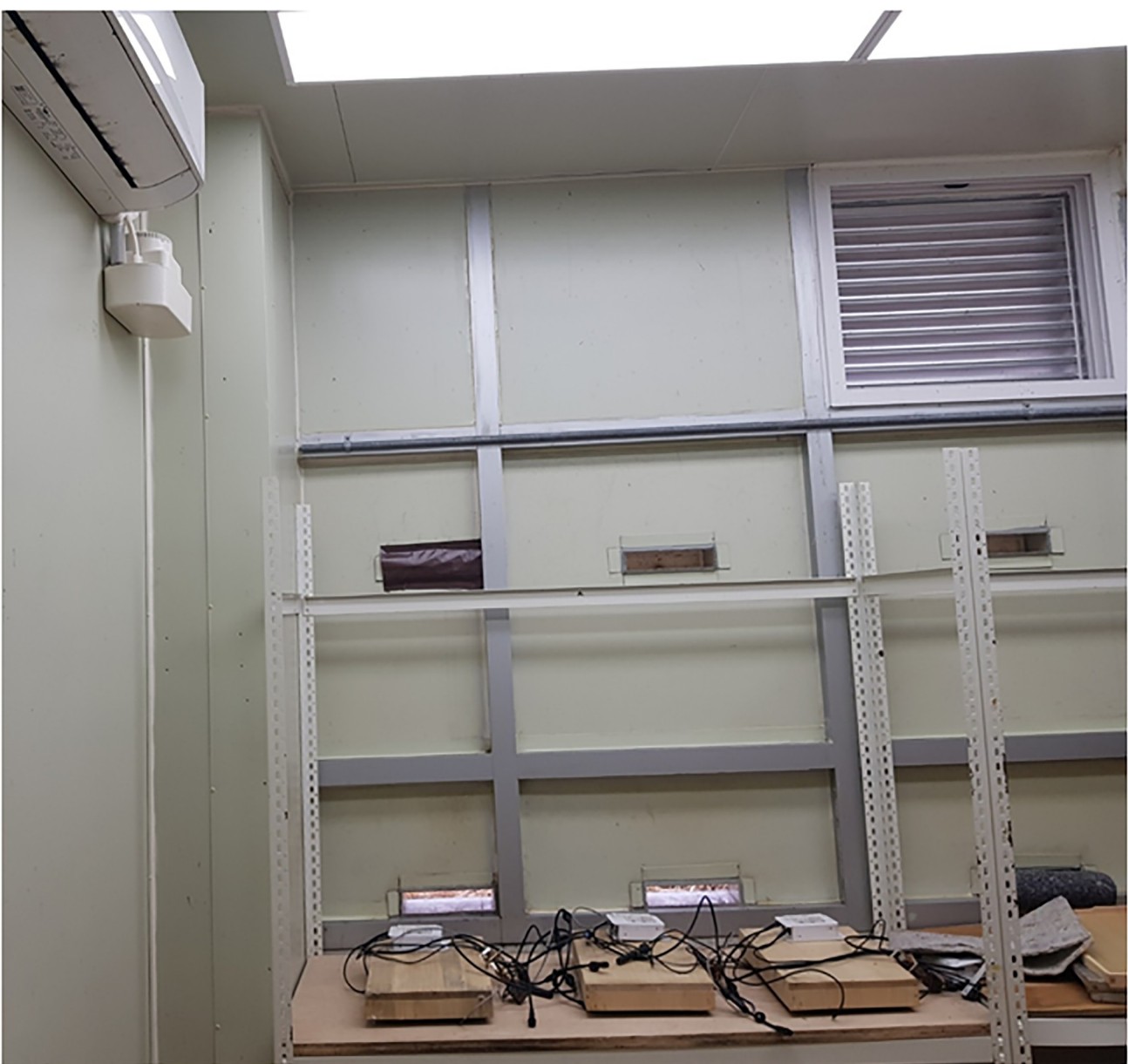

**Fig 1. The room environment before the experiment.**

Throughout summer, as well as in September and October of 2022, the honey bee colonies were subjected to ambient temperatures. Starting from November 2022, all colonies were kept under natural conditions and successfully overwintered. The success of colonies overwintering was monitored on February 5th and 27th, when we recorded whether the colonies survived (+) or did not survive (-) (S1 Table).

The overwintered marked bees, aged approximately 4.5 months, were sampled on February 5th, 2023, a time when queens had not yet begun laying eggs. Forty bees from each colony were sampled and individually placed in labeled 50 ml tubes, then stored at -80˚C before analysis.

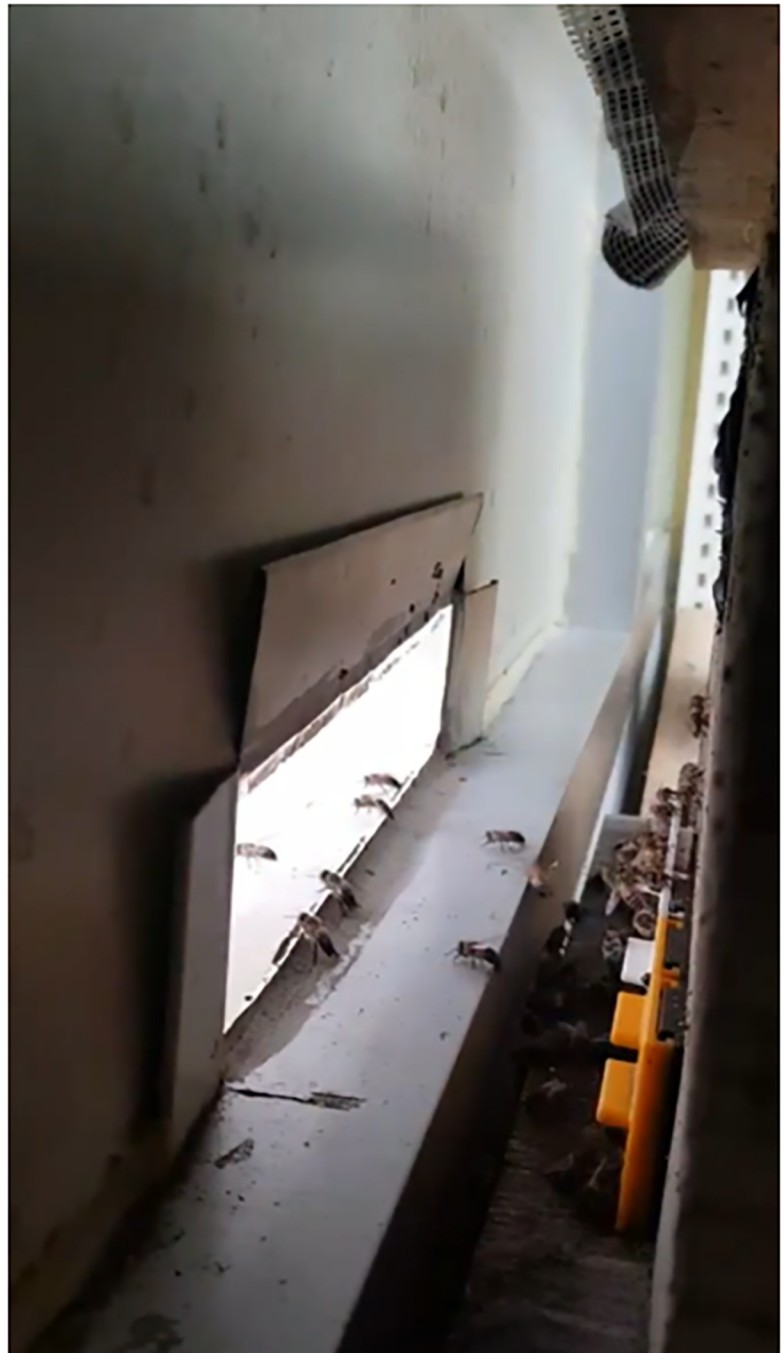

**Fig 2. The process of honey bees freely flying outside from the experimental room.**

## Brood microclimate measurements

Temperature, humidity (AM2320, BeeOnFarm, Korea), and $CO_2$ concentration (BO-100, BeeOnFarm, Korea) sensor systems were placed on the top and middle of each colony in August 2022. The data logging system was included in the sensor system because it provided a real-time clock. Data consisting of time (seconds), $CO_2$ (in ppm), temperature (in ˚C), and

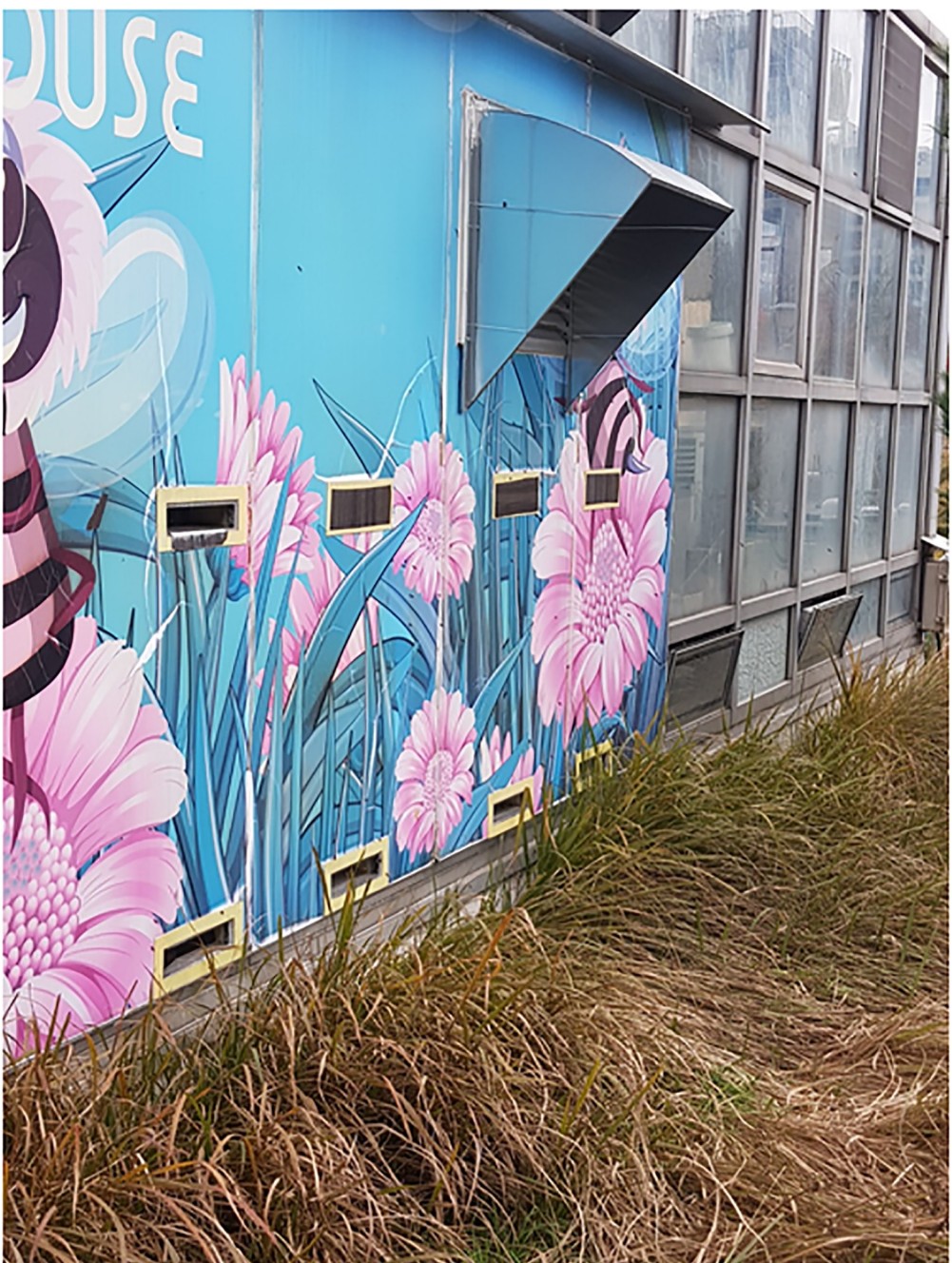

**Fig 3. The view of the room and the entrance for bees outside.**

relative humidity (as a percentage) were saved (S1 File). The data from 6 colonies kept in each room (three at 25˚C and three at 35˚C) were collected over 15 days from 11 to 25 August. Average values for the day were used for statistical calculations.

### Hypopharyngeal gland (HG) measurements

Acinus size from the hypopharyngeal glands of ninety overwintered bees in each Group (control, 25˚C, and 35˚C) were measured to determine if there were differences based on the ambient

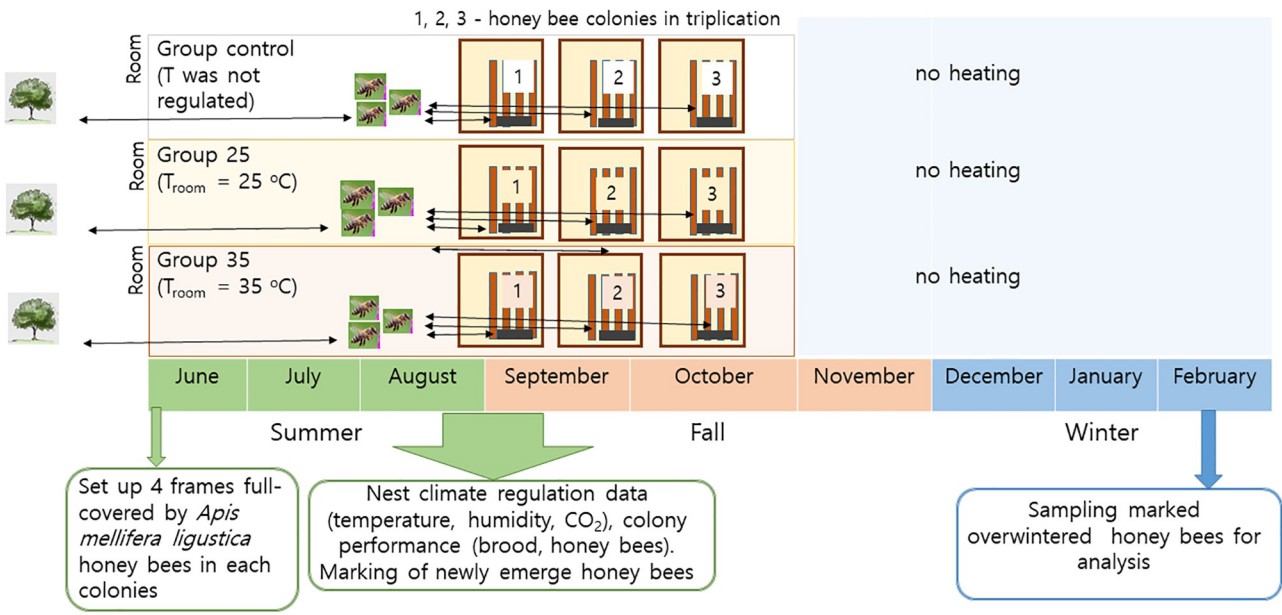

**Fig 4. Overview of the experimental design.** Timeline of the experiment.

temperature during the previous seasons. Thirty bees were collected from each colony, flash frozen in liquid nitrogen, and maintained at −80˚C until their glands were measured (S2 File). For each bee, the HGs were dissected into 1× phosphate-buffered saline (PBS) with 0.1% Tween$^®$ 20 and visualized at 40× magnification. The size (mm$^2$) of at least ten randomly selected acini per bee was measured using a Nikon Eclipse E200 microscope (Nikon, Tokyo, Japan) equipped with ToupView software (Touptek, Hangzhou, China) installed on a PC. Only acini with clearly defined borders were included. Acinus size was averaged within each individual.

## Fat body lipid content

The fat body lipid content in the abdomen of overwintered honey bees was studied in early February by the standard method [57, 58]. Ten bees per colony were collected; the ninety bees

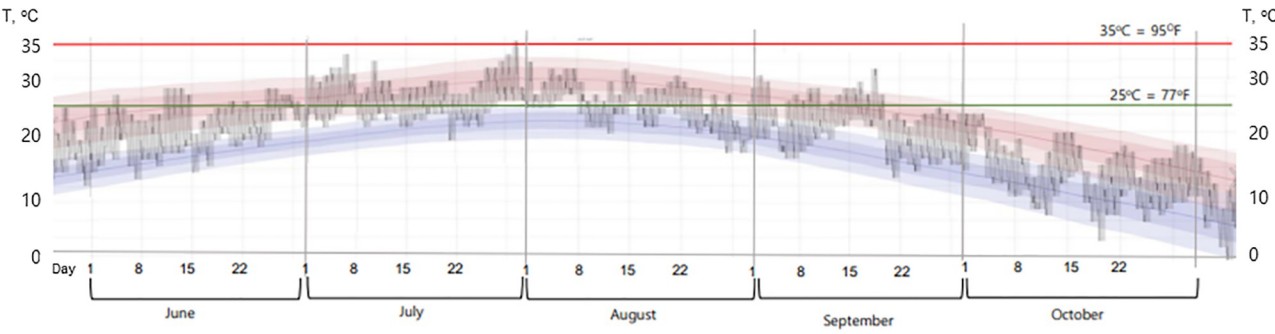

The daily range of reported temperatures (gray bars) and 24-hour highs (red ticks) and lows (blue ticks), placed over the daily average high (faint red line) and low (faint blue line) temperature, with 25th to 75th and 10th to 90th percentile bands.

**Fig 5. Ambient temperature of the group control, Group 25 (green line), and Group 35 (red line) under the temperature experiment in 2022, Incheon, Republic of Korea.** The data were obtained from https://weatherspark.com/h/m/142040/2022/10/Historical-Weather-in-October-2022-in-Incheon-South-Korea#Figures-Temperature.

were flash frozen in liquid nitrogen and maintained at −80°C until their abdomens were dissected and their guts were removed. Pools of ten abdomens per colony were weighed on a scale (HS220S, HANSANG Instrument Co., Ltd., Republic of Korea) before and after drying to a constant weight in a fume hood (SSFH-1000, Shinsaeng, Republic of Korea). The total lipid content was determined by measuring the decrease in weight following the removal of lipids from the dried tissue through extraction in 5 ml portions of diethyl ether (179272–1 L; Sigma–Aldrich, USA) (S2 File).

## RNA extraction and cDNA synthesis

A total of ninety bees (ten winter bees from each colony) were sampled in labeled tubes and stored at -80°C. The brain and gut-free abdomen of five sampled bees from each colony (fifteen bees per treatment) were dissected on ice, and RNA was extracted directly. RNA was extracted from the brain for the analysis of *HSP70*, *JHAMT*, *ilp1*, *ilp2*, and *TOR1* expression, and from the abdomen for *vg* gene expression analysis, as this gene is expressed in the fat body located in the abdomen. Genes are measured in the brain to understand their roles in stress responses (*HSP70*), metabolic regulation (*JHAMT* and *TOR*), and neural function (*ilp1* and *ilp2*). Total tissue RNA (brain or abdomen) was extracted using a Qiagen RNeasy Mini Kit (#74,104; Qiagen, Valencia, CA, USA). The total RNA concentration and purity were quantified using OD260/OD280 values between 1.8 and 2.0. Next, reverse transcription was performed using an RNA to cDNA EcoDryTM Premix (Oligo dT) kit (Takara, Japan). The reverse transcription reaction mixture included 50 ng/μl total RNA (with the clear volume calculated for each sample) and RNase-free water for a total volume of 20 μl. Reverse transcription was conducted at 42°C for 60 min, followed by heating at 70°C for 10 min.

## Quantitative real-time PCR

The relative expression of six genes (*vg*, *ilp1*, *ilp2*, *TOR1*, *JHAMT*, and *HSP70)* was measured. The housekeeping gene *RpL32* (Ribosomal protein L32) was used as an endogenous control. The PCR primer sequences are shown in S2 Table.

The reaction conditions were optimized. The qRT–PCR mixture had a volume of 20 μl which included 2 μl of template cDNA, 10 μl of Green qPCR Master Mix, 1 μl of upstream and downstream primers (5 pM/μl), and 6 μl of nuclease-free water (AM9930, Invitrogen, USA). Quantitative real-time PCR (qRT–PCR) was conducted using Brilliant III Ultra-Fast SYBR® Green qPCR Master Mix (600882, Agilent Technologies, USA) on an AriaMx Real-Time PCR System (Agilent Technologies LDA, Malaysia) with AriaMx 96 Well Plates, Skirted, LP (401490, Agilent Technologies, USA) covered by MicroAmp™ Optical Adhesive Film (4311971, Thermo Fisher Scientific, USA).

The qRT–PCR amplification procedure was as follows: initial denaturation at 95°C for 10 minutes; 40 cycles of denaturation at 95°C for 30 seconds, annealing at 60°C for 25 seconds, and extension at 72°C for 15 seconds. Each sample was replicated three times. Data analysis was performed using Agilent AriaMx version 2.0 analysis software. Relative gene expression data were analyzed using the 2^(-Delta Delta C(T)) method [59, 60] (S3 File).

Confirmation of amplicons in the RT–PCR products was performed by separating them through electrophoresis in a 2% agarose gel at 80 V for 40 minutes and analyzing them using a gel documentation system, the Gerix 1010 transilluminator (Biostep GmbH, USA) (S2 Fig). A Dyne 50 bp DNA Ladder (Cat. No. A701, DYNEBIO, Korea) was used as a reference.

| Month of sampling | | *Apis mellifera ligustica* honey bees | | |
| --- | --- | --- | --- | --- |
| | | Group control (3 colonies) | Group 25 (3 colonies) | Group 35 (3 colonies) |
| Experimental condition (summer, September, and October) | | | $T_{room}$ = 25 ºC | $T_{room}$ = 35 ºC |
| August September | Data set 1 (brood microclimate and colony development) | | Nest climate regulation | |
| | Statistics of data set 1 | | T-test ($p < 0.05$) | |
| February | Sampling of honey bees, n | (for physiology markers): 30+30+30 (for molecular markers): 10+10+10 | 30+30+30 10+10+10 | 30+30+30 10+10+10 |
| | Data set 2 | Two physiology markers (hypopharyngeal glands (HG, acinus size) and lipid content of fat body) Molecular marker genes related to nutrition (*vg, ilp1, ilp2, TOR1*), development (*JHAMT*), and stress response (*HSP70*). | | |
| | Statistics of data set 2 | 1. Discriminant analysis (define the variables with high discrimination score); 2. Agglomerative Hierarchical Clustering (check the natural groupings within dataset); 3. Elastic Net Regression (models the relationship between ambient temperatures, HG (acinus size) and lipid content of fat body with molecular markers); 4. ANOVA, Duncan's post-hoc test (comparison of significance within each variable, $p < 0.05$). | | |

**Fig 6. Experimental conditions, sampling schedule, data sets, and statistical tests used for analysis.**

## Statistical analysis

The statistical analysis, designed as illustrated in Fig 6, were conducted using Microsoft Excel and XLSTAT software (Addinsoft Pearson Edition 2014, Addinsoft, Paris, France). The characterization was performed using the following strategies [61]: (a) classification of the marker dataset to find a combination of features that best separates different groups (supervised approach, Discriminant Analysis (DA)); (b) clustering to discover natural groupings in the absence of labeled information (unsupervised approach, Agglomerative Hierarchical Clustering (AHC)); and (c) modeling the relationship between independent variables and a dependent variable and dealing with multicollinearity in the marker dataset to rank the markers by importance for identifying overwintered bee task development (supervised approach, Elastic Net Regression).

Specifically, the bees were categorized into groups, and the variables with the highest discrimination scores were selected using DA because it determines the linear combination of variables and provides maximal separation between groups. To discriminate the groups, Fisher distances were applied, which emphasize the linear combinations of variables that maximize the differences between groups.

AHC analysis provided a visual representation of the clustered dataset, revealing the arrangement of individual data points or observations based on their dissimilarity. The evaluation metric for clustering algorithms was the silhouette index. The score is bound between (− 1) for incorrect clustering and (+ 1) for highly dense clustering.

Elastic Net Regression analysis was used to test three models of the relationship [61, 62] between temperatures, physiology and molecular markers. The dependent variables were ambient temperatures, acinus size and lipid content in the fat body. The seven independent variables were ranked by coefficient. Receiver operating characteristic (ROC) curves were generated to quantify how accurately regression analysis can discriminate between honey bee groups [63]. The ROC specificity and sensitivity plot demonstrated a perfectly fitted model linking physiology to molecular markers (gene expression profile) (S1 Fig). The area under the curve (AUC) can range from 0.5 to 1.0, with a preference for higher scores, meaning that the model effectively distinguishes between the two classes.

The mean, standard deviation, and variance of each gene were calculated using descriptive statistics and visualized using the heat map module. For the gene expression analysis, ANOVA was used to test overall effects, followed by the Duncan's post hoc test ($p < 0.05$) for multiple comparisons and the t-test for differences of means between two groups of bees (S4–S6 Files).

## Results

The experiment was conducted on nine colonies of western honey bees (*Apis mellifera ligustica*). Each Group (control, 25˚C, and 35˚C) was represented by three colonies. The bees in Groups 25 and 35 experienced constant diurnal ambient temperatures (modeling global warming) from June to October, while the room for the Group control was freely ventilated to allow for natural ambient temperatures. All bees experienced natural foraging outside without restrictions. Newly emerged bees from each Group were marked in September for the overwintering experiment in February.

### Dynamics of ambient temperature in summer and fall months

The daily temperature in the first experimental room was set at 25˚C for five months, and there were no significant differences compared to the natural daily average temperature in the summer months or September (t-test, $p > 0.05$). The difference in experimental conditions between the Group control and the Group 25 was due to diurnal temperature fluctuations, which were eliminated in Group 25, where warmer days and cooler nights did not affect the bees. In our experiment, this stable elevated night temperature in Group 25 simulated a global warming scenario. However, the temperature in room Group 25 was significantly higher than the average daily temperature in October 2022 (t-test, $p < 0.0001$). The daily temperature in the second room was set at 35˚C for five months, which was significantly greater than the average diurnal temperatures during the summer months, September, and October (t-test, $p < 0.0001$).

### Brood microclimate and colony development in August and September

In this study, we hypothesized that the regulation of the nest climate by colony performance influences the physiology of winter bees and the success of overwintering. The temperature, humidity, $CO_2$, number of larvae, capped brood area, and numbers of adult bees from August 08 to September 18, 2022 were evaluated, when short-lived bees commenced rearing larvae that would develop into long-lived bees for the overwintering period. Since the expected ambient temperature in August and early September and the comparison of the colony performance of Group control and Group 25 on August 08 (Figs 7–9) showed no significant differences, sensors were not set up in the Group control.

   The mean and standard deviation of the brood temperature for bee colonies kept at 25˚C and 35˚C were 30.00 ± 0.98 and 33.40 ± 0.15˚C, respectively. Similarly, the brood humidities for these groups were 60.00 ± 0.43 and 64.72 ± 8.65%, and the brood $CO_2$ concentrations were 813.55 ± 296.80 and 633.30 ± 117.00 ppm, respectively. Importantly, the brood temperature inside the colonies kept at 35˚C was significantly greater than that inside the colonies kept at 25˚C (t-test, $p < 0.01$) (Fig 7). However, brood humidity, $CO_2$ concentration, number of larvae, capped brood cells, and number of bees were not significantly differing (Figs 8 and 9: t-test, $p > 0.05$) (Figs 10–12: ANOVA, Duncan's post hoc tests $p > 0.05$). The control colonies, however, exhibited significantly (ANOVA, $p < 0.01$) larger capped brood area, more larvae, and a higher number of bees (ANOVA, $p < 0.05$) on September 18 compared to colonies in Groups 25 and 35.

### Distinguishing honey bee groups through marker ranking

The overwintered bees were distinguished by their physiological state using a dataset comprising two established physiological markers (acinus size area and lipid content in the fat body) and six molecular markers (*vg*, *TOR1*, *JHAMT*, *ilp1*, *ilp2*, and *HSP70*).

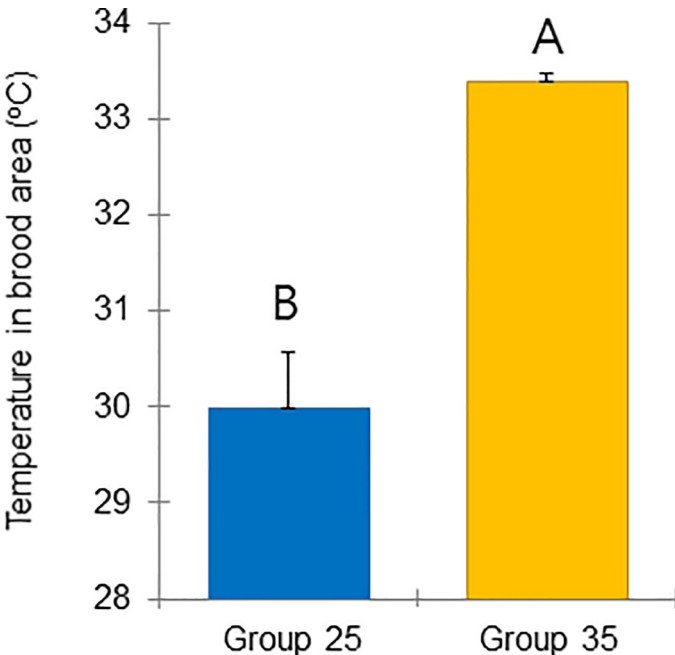

**Fig 7. Colony performance in honey bees treated under different ambient temperatures (08 August—18 September 2022).** Temperature in the brood area (°C). Statistical significance was determined using a t-test ($p < 0.01$, indicated by different letters above the columns) to compare the means of the two groups. SD–Standard Deviation, NS–No Significance. Honey bee colonies in Group 25 were maintained in room conditions at 25°C; colonies in Group 35 were kept in room conditions at 35°C; colonies from Group control were maintained under natural conditions.

Data Mining methods, including Discriminant Analysis (DA), Agglomerative Hierarchical Clustering (AHC), and Elastic Net Regression, were employed in the analysis.

DA was performed to plot the Groups control, 25 and 35, and to identify the optimal combination of features that effectively distinguishes them. The Wilcoxon Lambda test within the DA yielded a statistically significant result (two-tailed test, $p < 0.05$), indicating that the model successfully differentiated between groups (Fig 13). The observed Wilks' lambda value of 0.0 signifies a robust separation of the Groups control, 25, and 35, demonstrating a high level of discriminative power. A comparison revealed significant differences between Group 25 and Group 35 (F(2,8) = 19.37; two-tailed test, $p < 0.05$; Fig 13).

The Fisher distances (FDs) in the DA between the Groups control, 25, and 35 were significantly different (FD = 27511; $p < 0.05$ and FD = 19131; $p < 0.05$, respectively). Moreover, the FD density was also significantly different between the Group control and Group 25 (FD = 784; $p < 0.05$). The bees were separated along the primary discriminant axis F1 in 99.92% accuracy, where discriminant function coefficients for gene expressions *vg*, *HSP70*, and *JHAMT* were -20.09, -14.96, and 23.58, respectively. These coefficients were the highest, providing significant roles to this separation.

Subsequently, AHC was employed to identify natural groupings without labeled information using a dissimilarity measure. The evaluation metric for clustering algorithms, the Silhouette index between clusters 1 and 2, was 0.657, indicating successful clustering, albeit with moderate density (Figs 14 and 15).

Finally, Elastic Net Regression was used to rank the markers by importance by modeling the relationships between eight (Fig 16) or seven (Figs 17 and 18) independent variables and one dependent variable. This analysis was conducted separately for ambient temperature (Fig

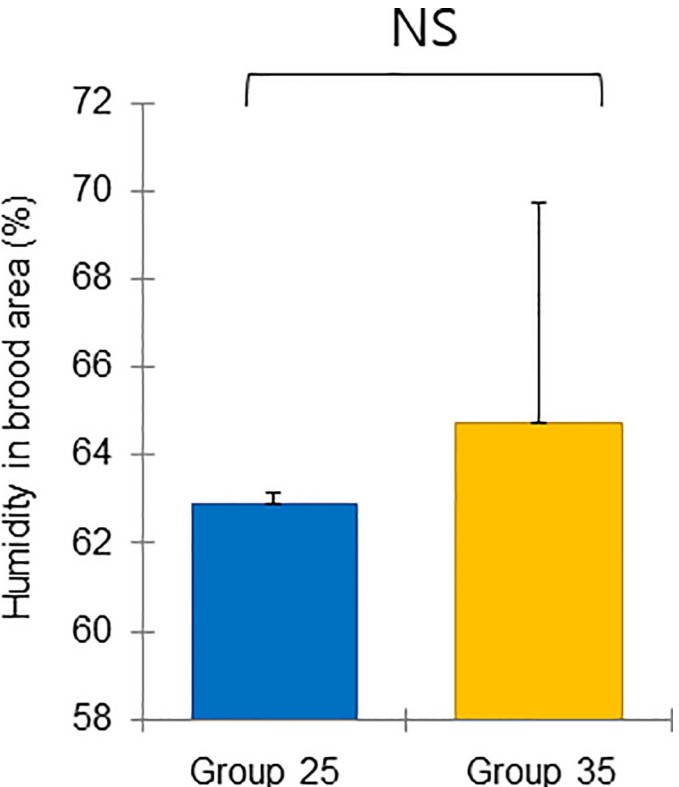

**Fig 8. Colony performance in honey bees treated under different ambient temperatures (08 August—18 September 2022).** Humidity in the brood area (%). Statistical significance was determined using a t-test ($p < 0.01$, indicated by different letters above the columns) to compare the means of the two groups.

16), fat body mass (Fig 17) and acinus size (Fig 18). The ambient temperature model identified significant predictors, with variable acinus size having the highest positive coefficient (305.773) and variable *TOR1* and *vg* having the highest negative coefficients (-29.969) and (-14.627) respectively. In the acinus size model, significant predictors were identified, with the variable *vg* having the highest positive coefficient (0.009) and the variable fat body mass having the highest negative coefficient (-0.009). In this all models *vg* showed high coefficient values.

## Physiological markers of honey bees

These results are based on the analysis of lipid content in the fat body and the acinus size of the hypopharyngeal glands (HGs), in overwintered bees. The mean and standard deviation of the lipid content for the bees in the Groups control, 25, and 35 were 4.33 ± 1.25, 4.00 ± 1.63, and 2.00 ± 0.82 mg/bee, respectively, with no significant differences (ANOVA, Duncan's post hoc, P = 0.22, $p > 0.05$), even though the lipid content in Group 35 was slightly lower (Fig 19).

Similarly, the mean and standard deviation of the size of the bee acinus from the Groups control, 25, and 35 were 0.026 ± 0.001, 0.027 ± 0.003, and 0.018 ± 0.001 mm$^2$, respectively. Notably, the acini in bees from Group 35 were significantly smaller (ANOVA, Duncan's post-hoc test, P = 0.006, $p < 0.01$) than in bees of the Group control and 25 (Fig 20). Importantly, both the lipid content and the acinus size of the overwintered bees in Group 35 were lower than those in Groups control and 25 (Figs 19 and 20), suggesting lower nutrition and different physiological state in bees from Group 35.

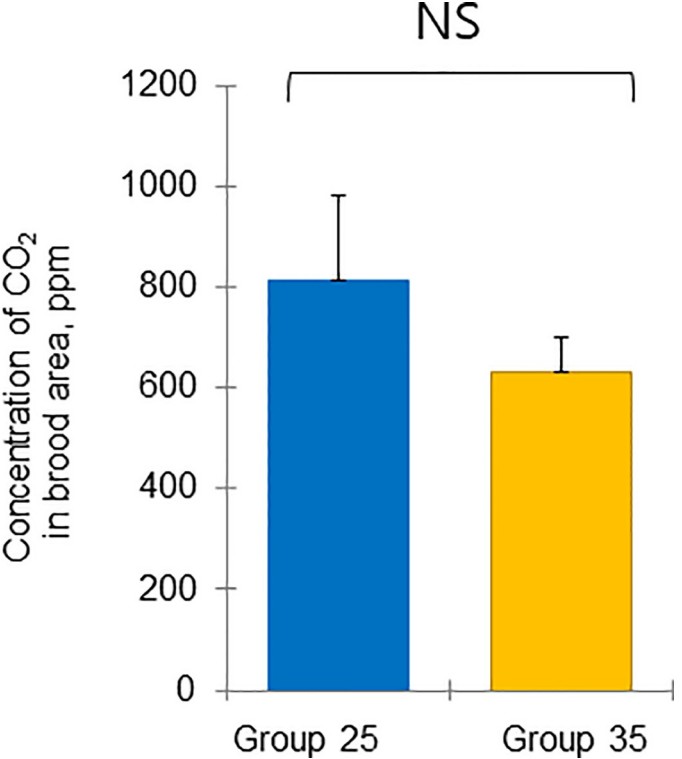

**Fig 9. Colony performance in honey bees treated under different ambient temperatures (08 August—18 September 2022).** Concentration of $CO_2$ in the brood area (ppm). Statistical significance was determined using a t-test ($p < 0.01$, indicated by different letters above the columns) to compare the means of the two groups.

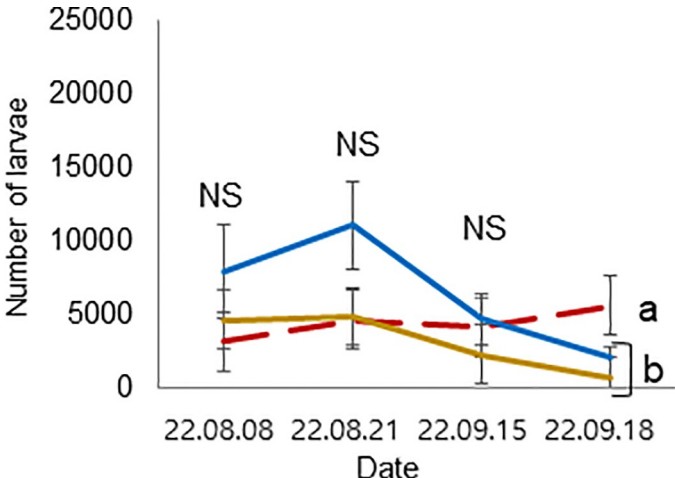

**Fig 10. Colony performance in honey bees treated under different ambient temperatures (08 August—18 September 2022).** Number of larvae. Statistical significance was determined using one-way ANOVA, Duncan's post hoc tests ($p < 0.05$, indicated by different letters above the columns) to compare the means of the three groups; bars in the graphs represent the mean ± SD.

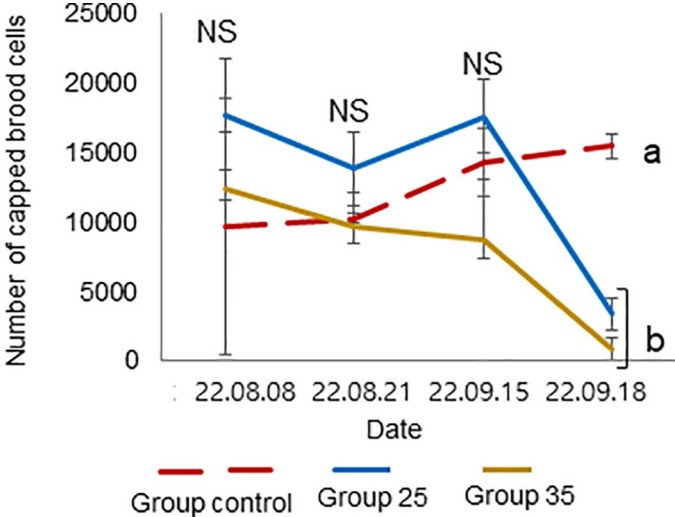

**Fig 11. Colony performance in honey bees treated under different ambient temperatures (08 August—18 September 2022).** Number of capped brood cells. Statistical significance was determined using one-way ANOVA, Duncan's post hoc tests ($p < 0.01$, indicated by different letters above the columns) to compare the means of the three groups; bars in the graphs represent the mean ± SD.

## Molecular markers of honey bees

These results include two previously defined task-related markers, *vg* and *JHAMT*, as well as four genes predicted to respond to physiology: *TOR1*, *ilp1*, *ilp2*, and *HSP70* (Figs 21–26).

Expression of *vg* gene was significantly upregulated (ANOVA, Duncan's post-hoc test, $p < 0.0001$) in bees from Group control and 25 compared to those from Group 35. In contrast, the expression of the *TOR1*, *JHAMT*, *ilp1*, *ilp2*, and *HSP70* genes was significantly downregulated (ANOVA, $p < 0.001$ for *TOR1*; $p < 0.0001$ for other listed genes) in the same bees, suggesting aging in bees from Group 35 compared to Groups control and 25 (Figs 22–26).

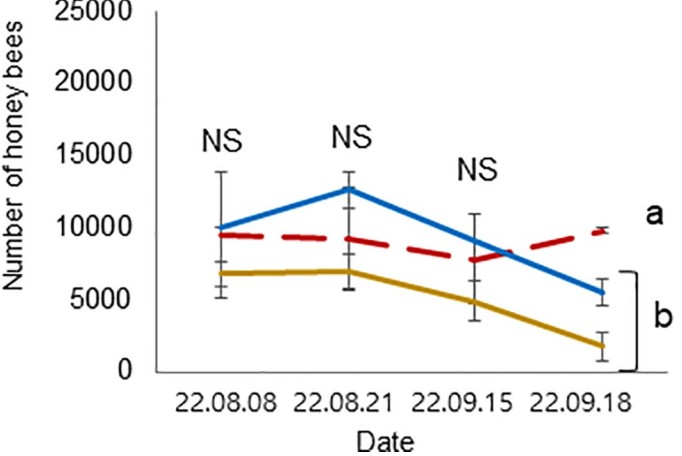

**Fig 12. Colony performance in honey bees treated under different ambient temperatures (08 August—18 September 2022).** Number of bees. Statistical significance was determined using a one-way ANOVA, Duncan's post hoc tests ($p < 0.05$, indicated by different letters above the columns) to compare the means of the three groups; bars in the graphs represent the mean ± SD.

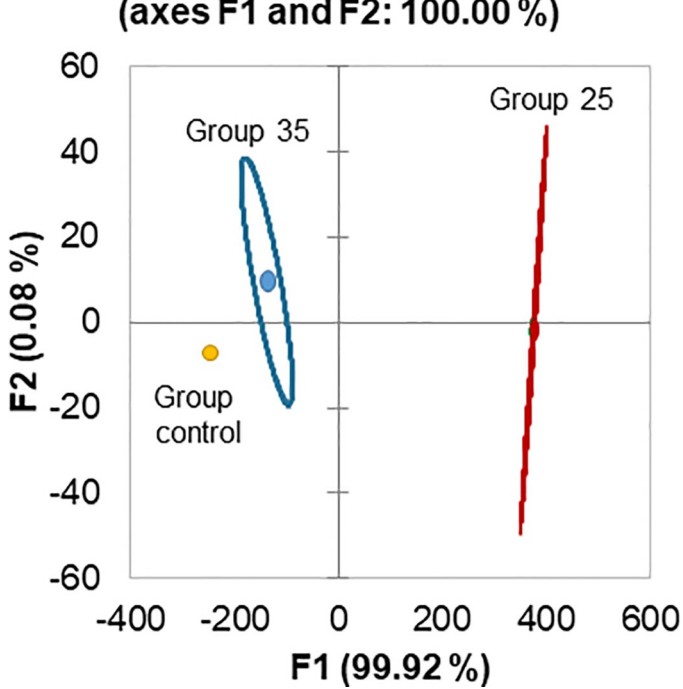

**Fig 13. The discrimination of overwintered honey bees was performed using eight variables related to aging: Two physiological variables (acinus size, n = 30; lipid content in the fat body, n = 10) and six molecular variables (n = 10), including the expression levels of *HSP70*, *ilp1*, *ilp2*, *JHAMT*, *TOR1*, and *vg* genes.** Discriminant Analysis (DA).

However, there were no significant differences in the expression of *TOR1*, *JHAMT*, and *ilp2* (ANOVA, $p > 0.05$) between bees from Group control and Group 25. The expression levels of *ilp1* and *HSP70* were significantly lower (ANOVA, $p < 0.05$) in the ones from Group control than in Group 25.

## Discussion

Key triggers influencing the transition of insects from summer to winter physiology include photoperiod, ambient temperature, brood microclimate, availability of pollen, and some yet-to-be-identified factors [25, 64–66], which obviously have a particular influence on overwintering physiological state, and also linked to aging and subsequent winter loss in bees.

However, due to the complex and unclear interactions among all these triggers, we designed an experiment in which only one factor associated with global warming—ambient temperature (25, 35°C, and a control)—was manipulated during the late summer, September, and October, and kept constant in experimental rooms. Colony development and brood climate regulation were investigated in August and September, during which long-lived (winter) bees were reared and marked to study the physiological state of bees alongside their clear chronological age in February. Physiology was assessed using two physiological markers (acinus size and lipid content in the fat body) and six molecular markers (*vg*, *TOR*, *JHAMT*, *ilp1*, *ilp2*, and *HSP70*) at the beginning of February, during the expected diapause continuation.

Since other colony parameters (e.g., food availability, hive size, and environmental conditions) excluding ambient temperature were consistent during the experiment (in summer and

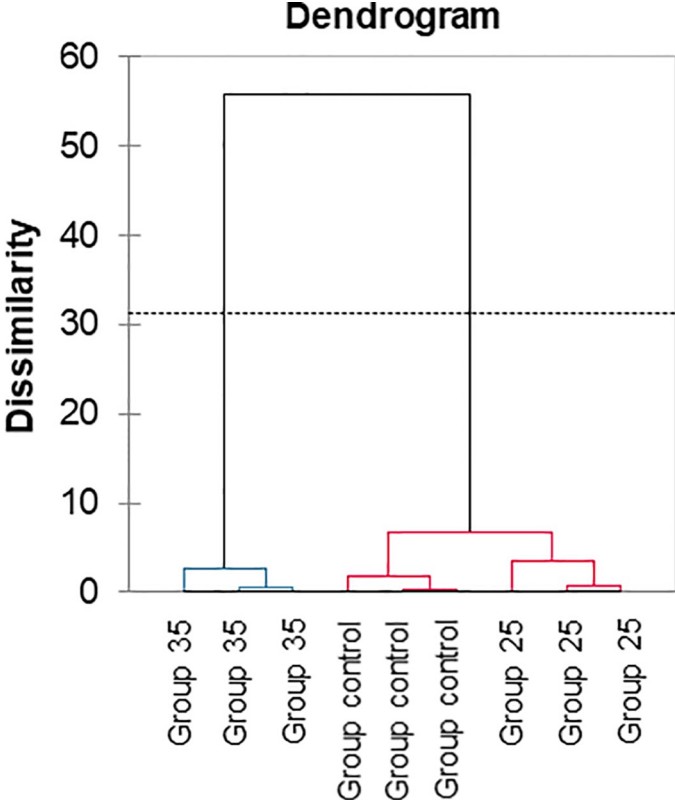

**Fig 14. The discrimination of overwintered honey bees was performed using eight variables related to aging: Two physiological variables (acinus size, n = 30; lipid content in the fat body, n = 10) and six molecular variables (n = 10), including the expression levels of *HSP70*, *ilp1*, *ilp2*, *JHAMT*, *TOR1*, and *vg* genes.** Agglomerative Hierarchical Clustering (AHC).

fall), the observed colony development in the Groups is likely due to the altered thermal environments, which can be the effects of temperature values and the constant diurnal temperature regime. It is crucial during ambient temperature extremes and heat waves that honey bee colonies rely on precise temperature regulation to maintain the brood microclimate [64] and support metabolic activity [4].

It was found that constant diurnal and seasonal temperatures (25˚C and 35˚C) significantly impaired colony development in Groups 25 and 35 in the fall season (from September 18 onward), resulting in reduced brood and worker numbers compared to the Group control. Also, brood microclimate was effectively regulated by the bees. Despite a 10˚C difference in the experimental ambient temperatures, their efforts reduced this difference to approximately 3˚C (Fig 7). Bees in Group 25 raised the brood temperature to about 30˚C, while those in Group 35 lowered the brood temperature to around 33˚C, slightly below the typical brood temperature of around 34.5˚C [67], reflecting well-documented microclimate regulation behaviors [21, 22].

Reduced brood and worker numbers, combined with challenges in brood temperature regulation, suggest that constant temperatures affect colony performance more than ambient temperature alone, as the average ambient temperatures for the Control and 25˚C groups in summer showed no significant difference. However, the brood temperature responses in Groups 25 and 35 suggest different survival strategies for the colonies. Although it is unclear

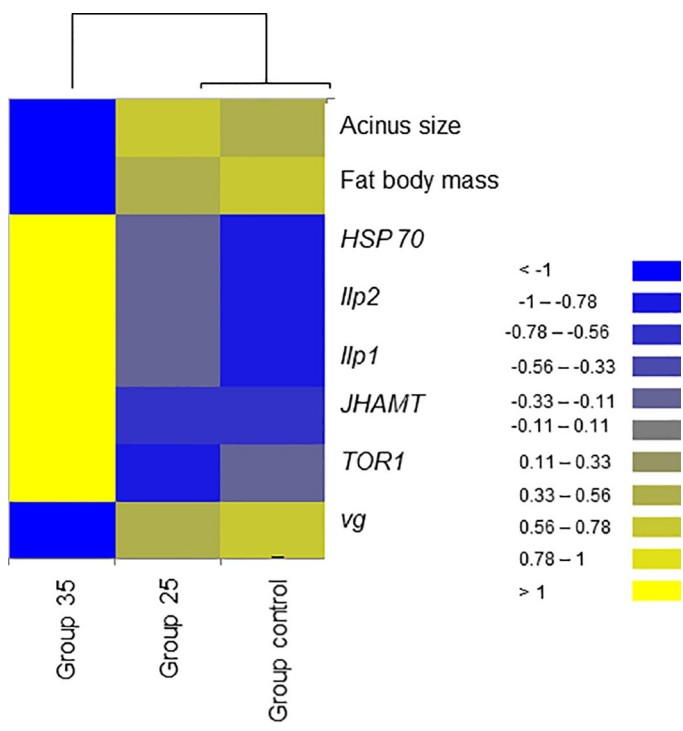

**Fig 15. The discrimination of overwintered honey bees was performed using eight variables related to aging: Two physiological variables (acinus size, n = 30; lipid content in the fat body, n = 10) and six molecular variables (n = 10), including the expression levels of *HSP70*, *ilp1*, *ilp2*, *JHAMT*, *TOR1*, and *vg* genes.** Heatmap with cluster analysis based on ddCt values, acinus size (mm²), and the lipid content in the fat body (mg).

whether the number of bees in colonies from Groups 25 or 35 was insufficient to heat or cool the hive, or if the subnormal brood temperature in these Groups were a strategy for rearing long-lived winter bees, these temperatures did not prove fatal for colony survival during over-wintering in our experiment (S1 Table). Furthermore, since Szentgyörgyi et al. [68] reported that short-lived bees tend to live longer after developing at a lower temperature (32°C), we favor this explanation. We hypothesize that reducing brood temperature may trigger physiological changes associated with extended longevity (long-live) in winter bees compared to summer bees [69].

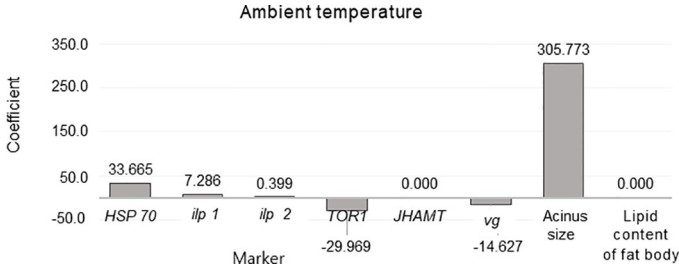

**Fig 16. Elastic Net Regression modeling the relationship between dependent variable ambient temperatures (settings in summer and fall) and the independent variables (*HSP70*, *ilp1*, *ilp2*, *TOR1*, *JHAMT*, *vg*, acinus size, and lipid content of fat body, in February) to rank the variables by importance for predicting dependent variables.**

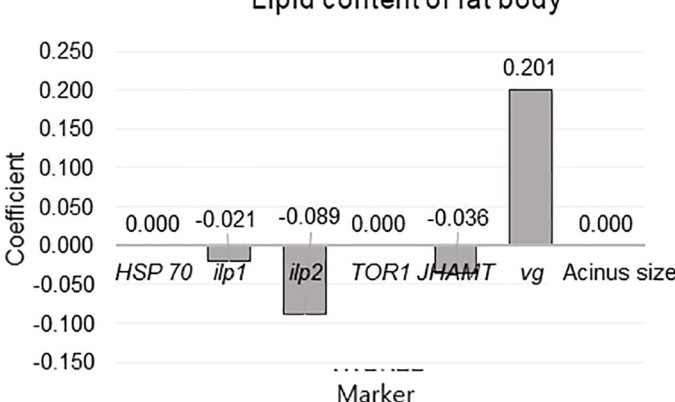

**Fig 17. Elastic Net Regression modeling the relationship between dependent variable lipid content in the fat body (in February) and the independent variables (*HSP70*, *ilp1*, *ilp2*, *TOR1*, *JHAMT*, *vg*, and acinus size, in February) to rank the variables by importance for predicting dependent variables.**

All these factors were expected to influence the physiological state of overwintered honey bees in diapause in February, which hatched under experimental temperatures. The known physiological markers of winter bees physiology include hypertrophied hypopharyngeal glands (HG) compared to newly emerged and forager summer bees [28]. This is important because the enlarged HGs of overwintering honey bees secrete more royal jelly to adequately feed the first larvae after diapause [39]. Thus, the nutrition of winter bees includes both the bee's own nutritional needs and the nutritional needs of larvae cared for by nurse bees, which seem to stimulate each other. Another marker is an enlarged fat body compared to summer bees [27], resulting from the accumulation of nutrients, particularly lipids [30]. These lipids support the production of hormones and molecular markers [40, 70, 71], as well as the synthesis of vg protein, which is later transported to the HG as a source of amino acids after diapause ends [72]. Based on this, larger acini in the HG and higher lipid content in the fat body of bees of the same chronological age may indicate a comparatively better physiological state. On February 5, winter bees from Group 25 and Group control, still in diapause, had the best physiological

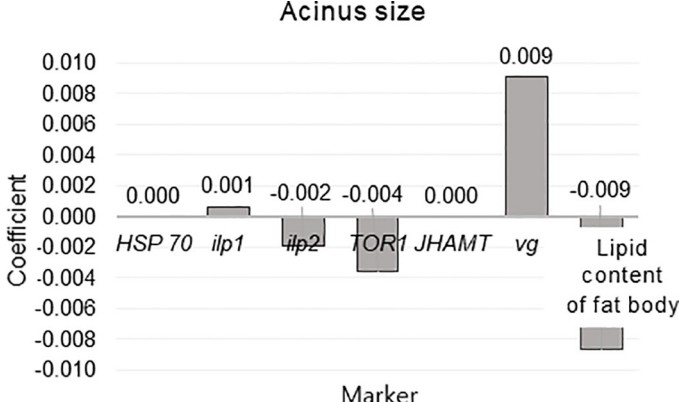

**Fig 18. Elastic Net Regression modeling the relationship between dependent variable acinus size (in February) and the independent variables (*HSP70*, *ilp1*, *ilp2*, *TOR1*, *JHAMT*, *vg*, and lipid content of fat body, in February) to rank the variables by importance for predicting dependent variables.**

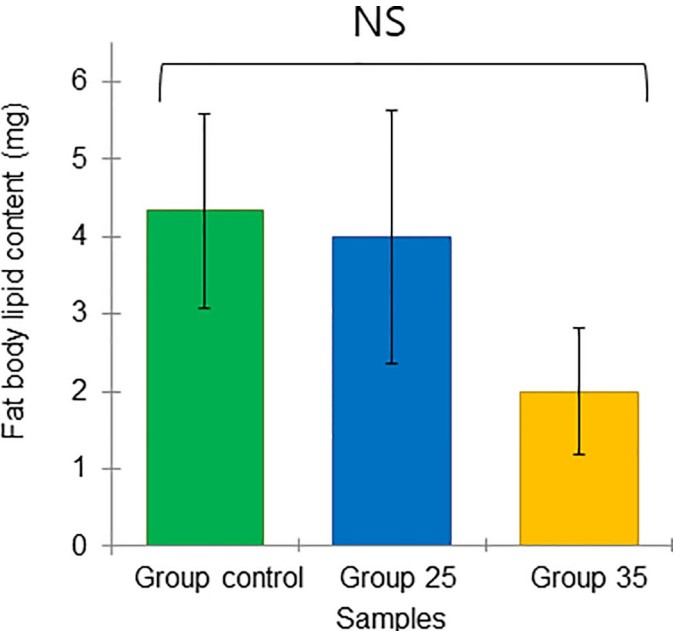

**Fig 19. Physiological marker of long-lived honey bees at the end of winter (February).** The average of lipid content in the fat body, n = 10 (mean ± SD). Statistical significance was determined using one-way ANOVA, Duncan's post hoc tests ($p < 0.05$, indicated by different letters above the columns).

state compared to bees from Group 35, as the acinus size in honey bees from Groups 25 and the Group control was significantly larger than in those from Group 35. A similar trend was observed in the fat body's lipid content, although the differences were not statistically significant.

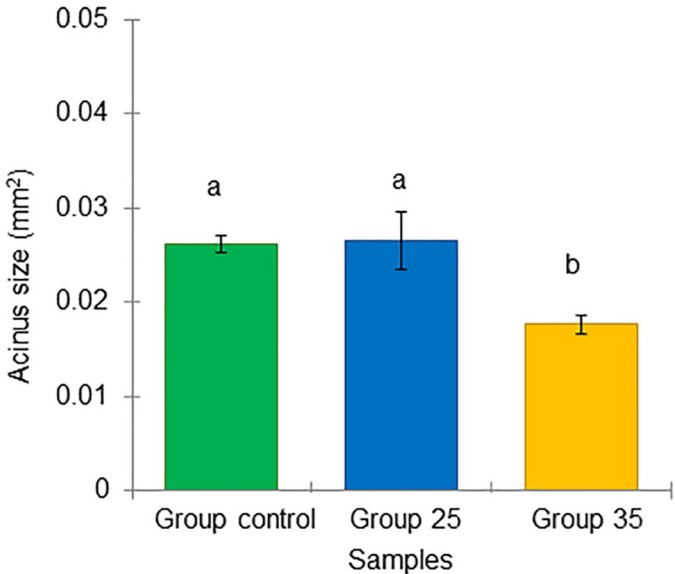

**Fig 20. Physiological marker of long-lived honey bees at the end of winter (February).** Average acinus size, n = 30 (mean ± SD). Statistical significance was determined using one-way ANOVA, Duncan's post hoc tests ($p < 0.05$, indicated by different letters above the columns).

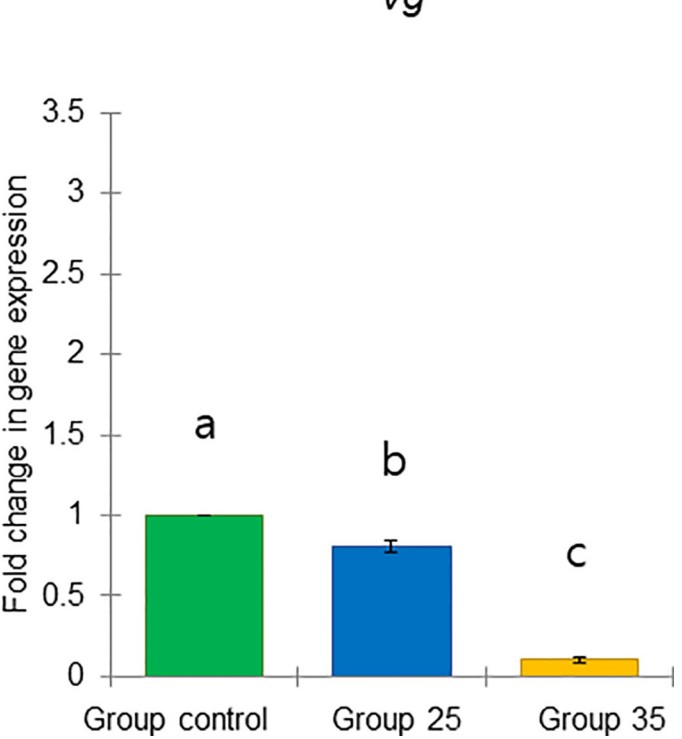

**Fig 21. The gene expression (ddCt) of *vg* in long-lived honey bees at the end of overwintering (February).**
Statistical significance was determined using one-way ANOVA, Duncan's post hoc tests ($p < 0.05$, indicated by different lowercase letters above the columns). The bars in the graphs represent the mean ± SD, n = 9.

The previously mentioned vg (gene and protein) is the most studied molecular marker in the dual suppression model in bees. Its elevation suppresses the levels of juvenile hormone (JH) and genes related to the JH pathway (*JHAMT*) [38]. This relationship has been used to compare the physiological states of both winter and summer bees [1, 33]. These molecular levels remain stable during diapause and undergo a switch before the onset of brood rearing [38]. Notably, *vg* gene expression was significantly higher in bees from Group 25 and the control, while *JHAMT* gene expression was significantly lower compared to Group 35. This indicates a different physiological state and suggests some changes in the aging process in the latter group of bees.

To utilize these well-known interactions, we built Elastic Net Regression models for ambient temperature, acinus size and lipid content in the fat body (Figs 16–18). In these models *vg* showed high negative (ambient temperature model) and positive (lipid content of fat body and acinus size models) coefficient values. This indicates new knowledge that high ambient temperature is related to *vg* levels in reared honey bees (Fig 16). It was also shown that during winter diapause, *vg* gene expression is highly related to acinus size (Fig 18) and lipid content (Fig 17) in bees under normal conditions. This supports the well-known relationship between *vg*, acinus size, and lipid content of fat body [14, 15, 27, 28, 37]. However, when each characteristic (*vg*, acinus size and lipid content of fat body) were analyzed separately, we found that in bees from Group control and Group 25, compared to Group 35, this relationship was reversed, suggesting potential physiological issues in bees from Group 35.

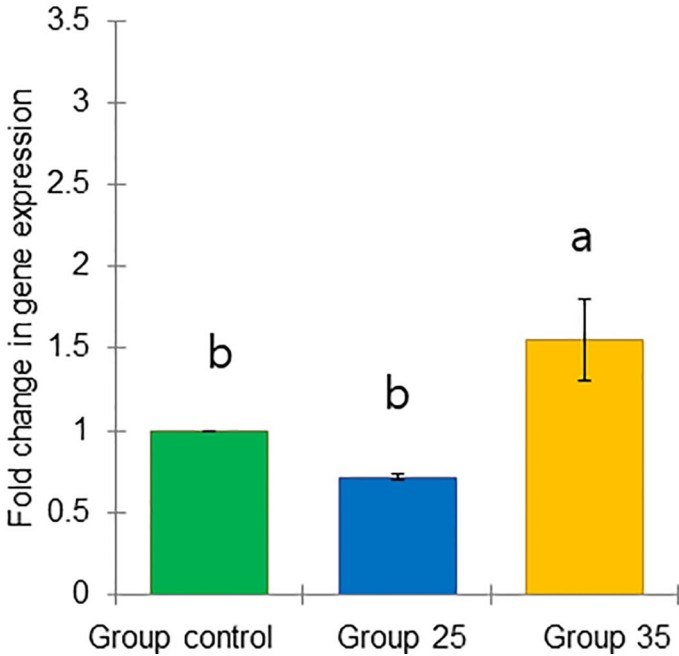

**Fig 22. The gene expression (ddCt) of *TOR1* in long-lived honey bees at the end of overwintering (February).** Statistical significance was determined using one-way ANOVA, Duncan's post hoc tests ($p < 0.05$, indicated by different lowercase letters above the columns). The bars in the graphs represent the mean ± SD, n = 15.

To identify genes that can significantly discriminate between the three treatment groups in our study, Discriminant Analysis (DA) was implemented. Three genes were identified with high scores: *JHAMT* and *vg*, which are well-known genes related to physiological states and are also suggested to be associated with aging, and the *HSP70* gene.

*HSP70* gene responds to stress [73] and diets along with *vg* in bees during winter [55]. We found the highest *HSP70* expression in bees from Group 35 and the lowest in the Group control, possibly due to stress related to their diet. We suggest that this may indicate increased nutritional needs at the end of diapause in bees from Group 35, as a significant increase in *TOR1* gene expression was observed in this group. This evolutionarily conserved gene integrates signals from nutrients (amino acids and energy) and growth factors to regulate cell growth [50, 74, 75]. The nutritional requirements of bees are supplied by foods stored in their nest (bee bread and honey) which influence their physiology. Winter bees in diapause require carbohydrate-rich food (honey) without a protein source, such as bee bread, to produce royal jelly related to *vg* metabolism, as there is no brood during diapause [17, 27]. This protein (vg) will be used after diapause ends, when the bees' HG increase in size to secrete royal jelly for feeding the larvae [1]. Additionally, significantly higher *HSP70* and *TOR1* gene expressions in bees from Group 35, compared to Group control and Group 25, were associated with a low in fat body lipid content, although it was not yet statistically significant. This supports the hypothesis that honey bees from Group 35 experienced long-term effects of ambient temperature, which are reflected in physiological changes indicated by two new molecular markers (high expression of the *HSP70* and *TOR1* genes), suggesting possible differences in diets and increased nutritional needs.

## JHAMT

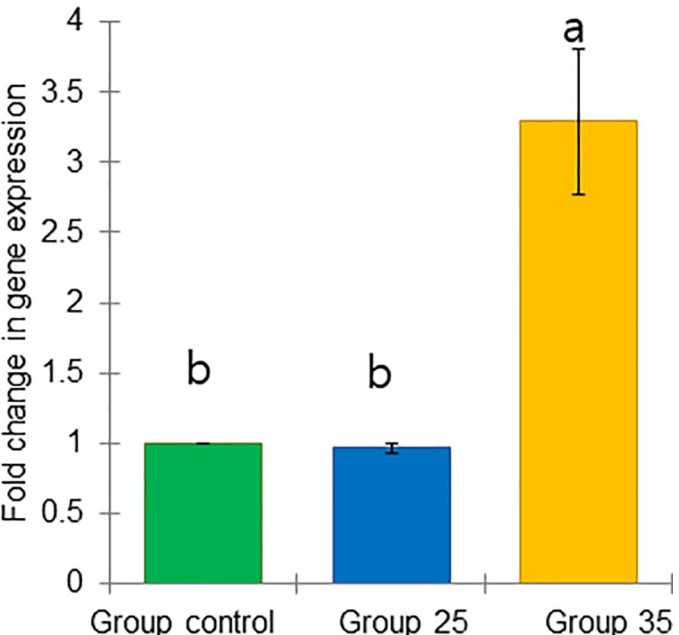

**Fig 23. The gene expression (ddCt) of *JHAMT* in long-lived honey bees at the end of overwintering (February).**
Statistical significance was determined using one-way ANOVA, Duncan's post hoc tests ($p < 0.05$, indicated by
different lowercase letters above the columns). The bars in the graphs represent the mean ± SD, n = 15.

The gene *ilp1* and *ilp2* did not show a clear effect in our experiment to ambient temperature, which may be due to several factors. First, the interaction of these genes with nutrition, age, and season may involve more subtle or context-specific metabolic pathways that were not fully captured under our experimental conditions [76]. Additionally, *ilp1* and *ilp2* could be influenced by other unmeasured variables that obscure their direct effects [77]. It is also possible that the genes influence on these factors is non-linear or dependent on specific thresholds, meaning a wider range of experimental conditions may be required to detect their effect. Further investigation is needed to disentangle these complex relationships and better understand how *ilp1* and *ilp2* function under varying biological contexts.

In conclusion, our results demonstrate that constant diurnal temperatures negatively impact colony development in summer and fall, reflecting broader issues related to global warming, such as reduced brood rearing, smaller worker populations, and increased brood temperature regulation demands. We hypothesize that reduced brood temperatures trigger physiological changes in bees that may be linked to extended longevity in winter. Notably, bees raised at the higher temperature (Group 35) exhibited distinct physiological states in bees at the end of diapause in February compared to those from Group 25 and the Group control. Comparative analysis revealed that overwintered bees from Group 35 had smaller acini, lower *vg*, and higher *JHAMT* gene expressions. Additionally, we identified two new molecular markers (high expression of *HSP70* and *TOR1* genes), which had increased expression in overwintered bees from Group 35 related to constant diurnal temperature in summer and fall.

While we cannot definitively conclude that bees from Group 35 faced suboptimal conditions, as the physiological changes did not result in fatal outcomes, further research is

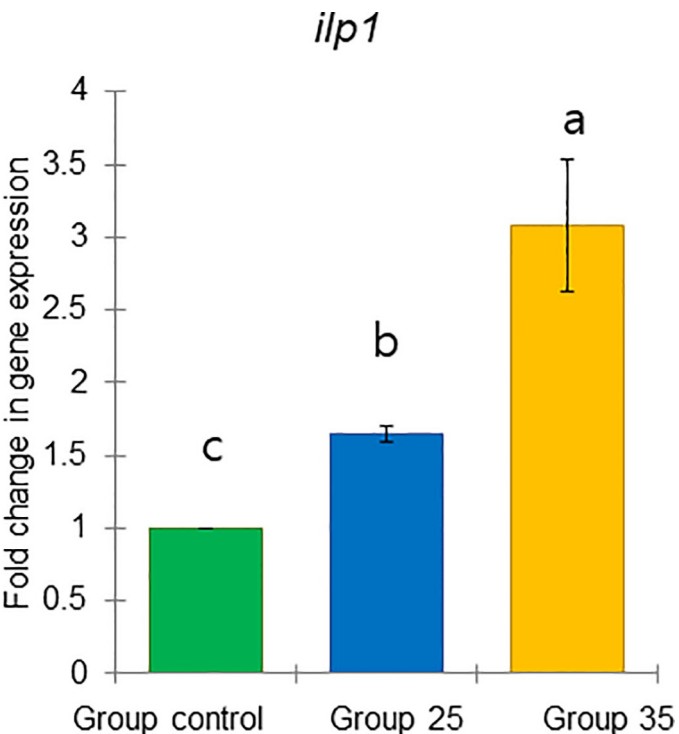

**Fig 24. The gene expression (ddCt) of *ilp1* in long-lived honey bees at the end of overwintering (February).**
Statistical significance was determined using one-way ANOVA, Duncan's post hoc tests ($p < 0.05$, indicated by different lowercase letters above the columns). The bars in the graphs represent the mean ± SD, n = 15.

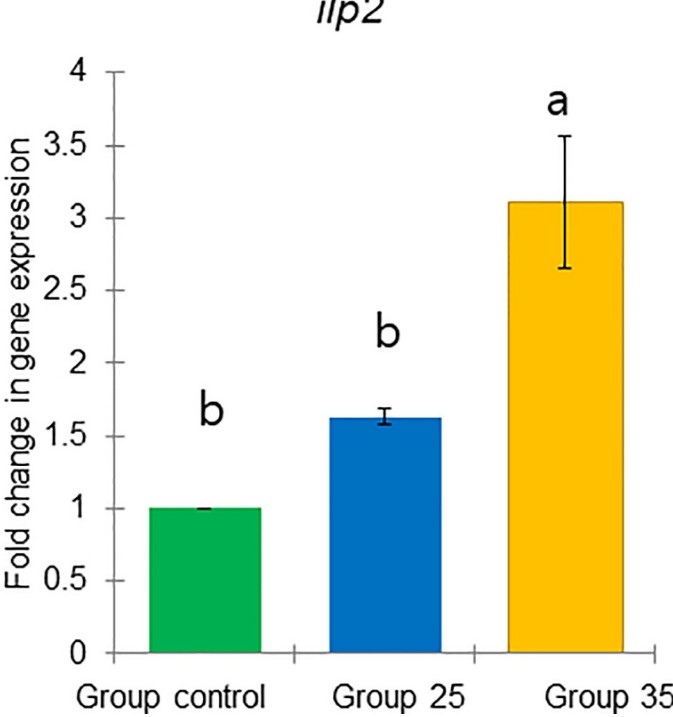

**Fig 25. The gene expression (ddCt) of *ilp2* in long-lived honey bees at the end of overwintering (February).**
Statistical significance was determined using one-way ANOVA, Duncan's post hoc tests ($p < 0.05$, indicated by different lowercase letters above the columns). The bars in the graphs represent the mean ± SD, n = 15.

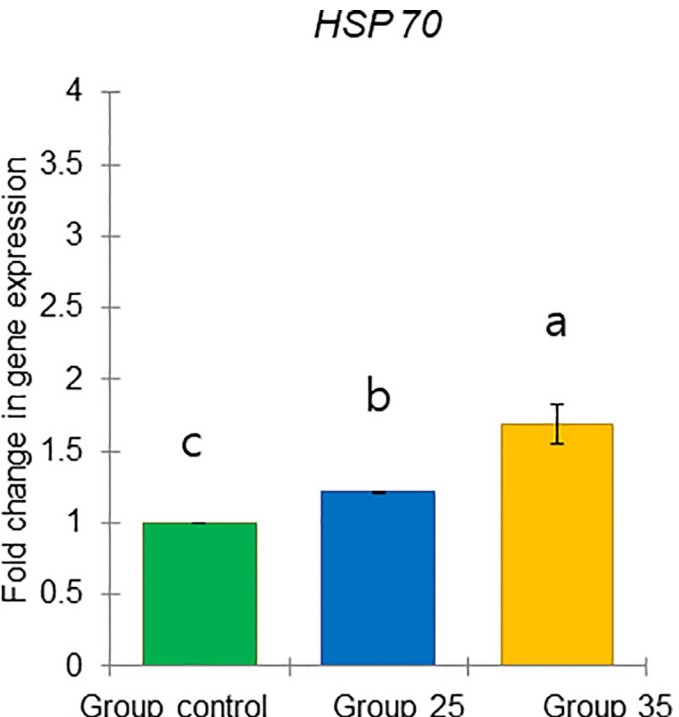

**Fig 26. The gene expression (ddCt) of *HSP70* in long-lived honey bees at the end of overwintering (February).** Statistical significance was determined using one-way ANOVA, Duncan's post hoc tests ($p < 0.05$, indicated by different lowercase letters above the columns). The bars in the graphs represent the mean ± SD, n = 15.

warranted. These findings enhance our understanding of how organisms respond to increased diurnal temperatures in summer and fall, providing valuable insights for apiculture practices and guidelines for winter management to prevent losses. Global warming during these seasons alters bee physiology for wintering, potentially threatening their longevity. Unlike bees experiencing typical fall conditions in Korea, which can adapt their colony development, those affected by global warming require assistance safeguarding their colonies at the end of winter and early spring.

## Supporting information

**S1 Table. Survival of honey bee colonies at the end of the overwintering.**
(PDF)

**S2 Table. PCR primer sequences.**
(PDF)

**S1 Fig. The quality verification plot (specificity vs sensitivity) of Elastic Net Regression model demonstrated a perfectly fitted model linking physiology to molecular markers (gene expression profile).**
(TIF)

**S2 Fig. Gel electrophoresis to verify the amplicons after qPCR.**
(TIF)

**S1 File. Raw data from sensors and XLSTAT 2022.**
(XLSX)

**S2 File. Raw data of acinus size, lipid content of the fat body, and colony performance.**
(XLSX)

**S3 File. Raw data qPCR.**
(XLSX)

**S4 File. Stat analysis in XLSTAT.**
(XLSX)

**S5 File. Raw data weather and statistics.**
(XLSX)

**S6 File. Raw data and stat colony development.**
(XLSX)

## Acknowledgments

We sincerely thank the Editor, Reviewer 1 and Reviewer 2 for their valuable and insightful comments, which greatly improved the quality of this manuscript. The authors thank BeeOn-farm company (Republic of Korea) for providing the equipment to record the in-hive micro-climate. Mention of trade names or commercial products in this publication is solely to provide specific information and does not imply recommendation or endorsement by the INU.

## Author Contributions

**Conceptualization:** Yumi Yun, Hyung-Wook Kwon.

**Data curation:** Yumi Yun.

**Formal analysis:** Olga Frunze, Hyunjee Kim, Ravil R. Garafutdinov.

**Investigation:** Young-Eun Na, Hyung-Wook Kwon.

**Methodology:** Olga Frunze, Ravil R. Garafutdinov.

**Resources:** Hyunjee Kim, Young-Eun Na, Hyung-Wook Kwon.

**Software:** Olga Frunze, Ravil R. Garafutdinov.

**Supervision:** Hyung-Wook Kwon.

**Visualization:** Olga Frunze.

**Writing – original draft:** Olga Frunze, Yumi Yun.

**Writing – review & editing:** Olga Frunze, Yumi Yun, Hyunjee Kim, Hyung-Wook Kwon.

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
