## [Decision Letter · Decision Letter 0]

16 Jul 2024

PONE-D-24-13353The effect of seasonal temperatures on the aging of the overwintered honey beePLOS ONE

Dear Dr. Kwon,

Thank you for submitting your manuscript to PLOS ONE. After careful consideration, we feel that it has merit but does not fully meet PLOS ONE’s publication criteria as it currently stands. Therefore, we invite you to submit a revised version of the manuscript that addresses the points raised during the review process.

Although quite complementary, the reviewers make many important points that I agree with and there concerns need to be thoroughly addressed.

We look forward to receiving your revised manuscript.

Kind regards,

Olav Rueppell

Academic Editor

PLOS ONE

Journal Requirements:

"Yes

Hyung Wook Kwon received.

This work was carried out with the support of the Cooperative Research Program for Agriculture Science & Technology Development (Project No. RS-2023-00232749) and the Priority Research Centers Program through the National Research Foundation of Korea (NRF) funded by the Ministry of Education (2020R1A6A1A03041954)."

Reviewers' comments:

Reviewer's Responses to Questions

**Comments to the Author**

1. Is the manuscript technically sound, and do the data support the conclusions?

Reviewer #1: Partly

Reviewer #2: Yes

2. Has the statistical analysis been performed appropriately and rigorously? 

Reviewer #1: Yes

Reviewer #2: Yes

3. Have the authors made all data underlying the findings in their manuscript fully available?

Reviewer #1: No

Reviewer #2: No

4. Is the manuscript presented in an intelligible fashion and written in standard English?

Reviewer #1: No

Reviewer #2: Yes

5. Review Comments to the Author

Reviewer #1: Comments on PLOS ONE ms. “Winter honey bees”

This manuscript describes an experiment designed to determine how elevated brood nest temperature affects the physiology and gene expression of wintering honey bees. I think the design of the experiment and statistical analyses were appropriate. However, I have substantial concerns about some of the wording and failure to address some of the implications of the results. Below I highlight some of my most important concerns. I have placed MANY comments and suggested changes to wording on the manuscript itself.

Despite my many comments, I enjoyed the study. I feel that the authors need to fix some of the glaring issues (i.e. HP glands are relatively undeveloped in wintering bees, not hypertrophied/well developed, as was mentioned at least 3 times), improve the wording, and better discuss the implications of their results with respect to our warming planet.

1. I anticipated that this study would lead to a discussion of how climate change, and specifically warming temperatures, may affect wintering bees. You mention this briefly in the last sentence of the Introduction. Therefore, I thought this would be the main point brought out in the Discussion. Instead, the Introduction and Discussion largely ignored the possible effects of climate warming on individual wintering bees. Why?

2. Lines 126-132 verbally describe the ambient temperatures for the three experimental groups. The Control and 25-degree colonies experienced similar temperatures, but experienced significantly lower temperatures than the colonies in the room kept at 35 degrees. The authors refer to Fig. 6E for these results—but there is no Fig. 6E. Moreover, Fig. 6E is referred to before Fig 1 in the text—figures must be referred to sequentially (1, 2, 3, …).

Related to this, the next paragraph describes how the brood nest temperature was significantly elevated in the 35 degree group compared to the 25 degree group. However, lines 141-149 of the Results fails to mention the control group! Figs. 1A 1B, 1C should include the control group. Figure 1E, that compares the numbers of capped brood cells over time, is therefore an enigma—as the 25 and 35 degree groups have similar numbers, but different from the Control, but the Control and 25 degree groups experienced similar temperatures in their experimental settings (except in the month of October). This odd result is never mentioned or discussed.

3. Line 48-49: You state: “it is necessary to elucidate the processes involved

in the transition from short-lived workers to long-lived ones.” This is a valid point, but it is not the focus of your research! Therefore, it creates an expectation in the reader of what you investigated, but later you explain that you are really interested in factors operating in fall that affect the longevity of workers during winter.

4. It is not clear why you chose to capitalize “Acinus” and “Fat body” throughout the manuscript. I see no reason to do so.

5. The use of the term “honey bee” throughout the paper. It is clear from the Introduction (lines 103-104) of the paper that you studied honey bees, Apis mellifera. You may want to state that again at the beginning of the Methods and Results, but in many many locations you can replace “honey bee” with “bee” or “bees”, or even in some places to reduce repetition with “it” or “them” (singular/plural).

6. “Acinus” is singular. In honey bees, an acinus is a single globular structure of the HP gland. The hypopharyngeal gland is composed of many grape-line “acini”- plural. You need to check through the paper to determine which mentions of “Acinus” should remain “acinus” (singular), and which mentions refer to multiple structures and should be referred to as “acini” (plural).

7. Line 63-64: The authors have written that the winter bee phenotype is characterized by “increased Acinus size (hypopharyngeal gland, HP)”. This statement is false. Refer to Fig. 2 in Doke et al., a paper you cited. Similar incorrect statements about larger HP glands in winter bees appears in Line 249 and Line 256.

You may also want to cite:

Wang K., Liu Z.-G., Pang Q., Zhang W.-W., Chen X.-M., Fan R.-L., et al. (2018). Investigating the regulation of hypopharyngeal gland activity in honeybees (Apis mellifera carnica) under overwintering conditions via morphologic analysis combined with iTRAQ-Based comparative proteomics. Ann. Entomol. Soc. America 111 (3), 127–135. 10.1093/aesa/say012

8. Lines 116-121: I feel that these sentences provide too much of the results. In the Introduction it is OK to briefly mention what the results are, in general terms, but this level of explanation of results seems excessive.

9. Statistical reporting: I have always referred to “Duncan’s post hoc test”. Not “Duncan”. I did an internet search and several sites also refer use the possessive noun: “Duncan’s”.

It is good to explain that SD refer to Standard Deviation the first time you use it in the text and in a figure legend. I do not think you need to write out “SD - Standard Deviation” repeatedly in the text and figure legends.

10. Number of significant digits to report: There are rules for how many significant digits one should report. It depends on the accuracy of the measurements as well as the number of possible values for the variable. Reporting mean temperature as 29.995 ± 0.982 °C is not correct. This should be either 30.0 °C or 30.00 °C. Please seek advice on significant digits and then check and correct all values reported in the manuscript.

11. Check the definitions of “weight” vs. “mass”. I believe you will find that you should refer to “fat body mass”.

12. Lines 152-154: Interpretations of data belong in the Discussion—not in the Results.

13. Line 155: Heading: “Molecular marker recognition in overwintered western honey bees”. I believe you determined “gene expression of molecular markers” And acinus size and fat body mass are not “molecular markers”. You need to reword this heading.

14. Figure 2: Because red-green color blindness is very common, particularly in men of European descent, one should never use red and green together in figures. This red and green graph will NOT be comprehensible by a number of readers. A different color scheme is needed.

15. Line 247: You state that the bees in Groups 25 and 35 successfully overwintered. I have seen no data in the manuscript that support that statement. There had to be some bee mortality over the 3.5 months. What percentage of marked bees survived to the date when they were sampled? Did you collect that data? (You may be referring here to colony survival, but that is not as relevant to your study as individual bee survival.)

16. Lines 247-248: I have placed two important comments on the manuscript related to this sentence.

Reviewer #2: Review "The effect of seasonal temperatures on the aging of the overwintered honey bee"

Frunze et al. investigate the effect of winter temperature experienced by larvae on aging markers in workers. To do so, hives are kept at two different temperatures plus a control treatment (natural conditions) over summer and fall. In winter workers of the hives of the different treatments are collected, fatbody size, acinus size and the expression of several aging associated candidate genes are investigated to make inferences on the putative effect of climate warming on aging in winter bees. The aim and topic of this experiment are highly relevant and interesting. At the moment, there are several parts and aspects that need clarification and correction. The authors should also discuss the interplay and the meaning of the various factors, from treatment to measured variables, more clearly.

Major comments

-I understand that you collected several candidate genes that have been shown to be associated with aging and you test these in the fatbody. It does not become clear in the discussion whether all these candidates change their expression in the expected direction. Do they differ and if yes how, and what is your interpretation of that?

-Why did you choose to investigate Vg gene expression in the fatbody, and the expression of all other candidates in the head?

-Why did you measure fatbody and acinus size?

=> the latter three points could be made clear in the introduction, but latest in the results

-What was the mean temperature of the control treatment in the hive, and how does it relate to the two treatment temperatures?

-The treatment temperatures differ by 10C, however, the in-hive difference is only 3C. This should be mentioned and discussed somewhere, also in respect to the control. This is especially relevant to be able to interpret why gene expression was only different to the 35C treatment.

-I find it highly interesting that the variables measured for Fig1D-F differ between the control and both treatments but later gene expression differs between 35C and the other two. Add this finding to the discussion and add your explanation for this pattern.

-Results of the statistical test need to be better depicted and also mentioned in full in the text.

See more details below.

Minor comments

Abstract

rephrase first sentence.

Line 23: Be more concise. Temperature is not the only factor reported to affect winter bee longevity, it would thus be good to at least write that it is ONE factor affecting longevity and thereby that other factors may also play a role. Or also mention other factors putatively affecting longevity in bee workers.

Be more specific in the abstract on what was in the treatment and who was tested later

Introduction

Line 41: maybe put emphasis first on their role as pollinators and honey production and then production of health and medical products.

Line 45: be more specific and add information on the "various factors" that may contribute to winter mortality. Plus maybe indicate that there are factors that contribute to strong or abnormal winter mortality. Because even under normal conditions winter bees will die and be replaced by spring workers...

Line 53: delete "seems"

Line 66/67: what do you mean with "unlike"? Do other insects need diet at all or other dietary composition?

Line 73/74: "Acinus" and "Fat" should be written in lower letters.

Line 77: what do you mean by "behavioral activities"? Why "inevitably" leading to reduced longevity and increased aging?

Please also add information on the roles and functions of JH and Vg as you do for the other candidate genes.

Line 96: "longevity" please be careful with using aging and longevity interchangeably. Longevity is basically the end result of the aging process, they are highly related but not the same.

Line 102: you might want to add some information on why and how temperature may affect lifespan.

Line 107: What exactly does "maintain" mean? Was the control group located outdoors or were outdoor conditions "simulated".

Line 117: "winter worker broods" unclear whether you investigated workers or brood. So far I understood that the aim was to investigate workers but here it sounds like it was brood? But then for Group35 you only talk about workers... Please phrase more clearly. => after reading the manuscript I now understand what you mean and it makes sense. Maybe consider writing "brood resulting in winter workers" or something like that?

Line 120: HSP70 and TOR1 may serve as additional markers for what? Please clarify. And what kind of research is additionally needed?

Line 122: I assume you hint towards faster aging under higher temperatures, but this is nowhere clearly stated. Please add this information and make the relationship easier understandable for the readers.

Results

since M&M comes at the end of the manuscript it would be helpful to get some additional information in the results already. E.g. at which stage you put the frames of which staged brood in the experimental conditions, you marked workers and those were tested later,...

Line 128 and all following statistical results: please add test statistics in addition to the p-value. Currently unclear which test you conducted and its effect size.

Line 132 and 312: Why do you start with Figure 6? This should be Figure 1.

Line 142: 29.995 ± 0.982 and 33.398 => the difference for the brood between the groups was only 3C instead of the 10C of the chambers. This should be mentioned much earlier. It basically means that the bees heat up the brood area to ~30C. Does 33C mean that the bees cool down the nest in the 35C treatment or are discrepancies in the ways nest and room temperatures are measured?

What were the according measures in the control group?

Line 156++: "The molecular markers recognition related to the biological age of long-lived honey bees, which were approximately 4.5 months old, was verified through a dataset comprising": sorry but I don't understand what you want to say here... E.g. what do the molecular markers "recognize"? And what was verified?

Line 168:"A comparison revealed significant differences between Group 25 and Group 35" Please be more specific. What kind of comparison, which statistics are you talking about here?"

Line 169: How many individuals were tested per colony. Based on Fig 2B I assume that 3 colonies per treatment were tested? One individual per colony or multiple? Please also add this information somewhere in the results, to give an idea of sample size. It would also be more telling to plot all the data points that went in the PCA (Fig 2A), which would also give an impression on the variation between samples.

Line 174+175: Please correct the interpretation of the DA results here: 99.98% is the percent variation explained along this axis. You could check based on the factor loadings which of the factors is the most telling here.

How does this figure indicate that gene expression of ... contribute to this separation? Please clarify.

Fig3: maybe move panel A to supplement. Panel B, add "fatbody size" to header.

Currently it is not clear what exactly you are testing. I.e. instead of "dependent variable" specify that you are testing which of the gene expression levels correlates with fatbody and acinus size. Also add that you are testing

I am actually wondering why you are conducting this analysis at all? If I understand correctly you are testing which of the variables, i.e. the expression of the different genes affects fatbody or acinus size, and whether fatbody size affects acinus size and vise versa. A) do you really expect that there is a causal relationship? B) I thought that the main aim was to test whether there are gene expression differences of these candidate genes between the different treatment groups, i.e. the results given in Fig 5.

Fig 4: again it would be better to depict the data as boxplot with single data points visible.

Line 207: change to "physiological aging"

Line 209: I understand what you mean with "physiological age" above, and it makes sense. Here I am lost what you mean by "biological age"? Is this also supposed to mean "physiological age", because based on M&M all the tested workers should be 4.5 months old and age matched, right?

Fig5: which measure for "relative expression level" did you use, deltaCT? Add information to Y-axes. Add measures to the Y-axes. Again depict results as boxplots with data points.

General comment in respect to results. P-values are mostly > or <0.05. "<0.05 means between 0.01-0.05, is this always the case, are p-values always in this range?

Discussion

Line 234: "during the previous summer" does this mean "during larval development"? Maybe specify this information for better understanding.

Line 243: you need to give the actual mean temperatures here as well, not just the difference. Otherwise readers might not know and/or remember that they had 30 C.

Line 248: what do you mean by "initially had to decrease the nest temperature"? It sounds like both groups had to...? But the 25 C group actually needs to increase temperature to reach 30C.

Please clarify that Vg and JH expression are connected. Which way and in which bees/castes should become clearer and to what extent does it fit to your results...

Line 276: "relationship between vg gene expression and the hypopharyngeal gland (HG) and Fat body" I neither see anything about the hypopharyngeal gland in Fig3, nor anywhere in the results, nor any relationship of this gland and Vg and the fatbody.

Line 282: " it is associated with honey bee transitions in summer and aging" not sure what you mean here? Please be more specific and clarify.

Line 286: "The subsequent gene, TOR1, wasn't selected by Data Mining methods but can be considered aging-related (Barth et al., 2010;Fahrbach et al., 2012). " This statement needs to be reformulated. You do not find an association of TOR1 expression with physiological aging in your study. There are other studies which do. What could be the difference why they find it you don't?

Line 288: "explaining the decrease in Fat body " this is not correct. You find a correlation of both not a causal relationship, please reformulate.

Line 293: this summary sentence is rather confusing.

Line 295: why "conversely"?

Did you continue observing those colonies? Did the differences in the 35C group lead to lower survival of the colonies, smaller colony sizes,... any fitness differences?

Figure 6D: change "no regulated" to "not regulated"

Line 357: change "whether differed" to "whether it differed"

Line 372: "name, company, country" add according information

Are the results of the lipid content given in the results, or somewhere in the manuscript? => either add results or remove information from M&M

Line 377: add information on how and when the tissue(s) were dissected. Before freezing at -80C? Were they dissected and then directly isolated? Were samples dissected on ice?

Line 435: change "group" to "groups"

Please indicate which tests were conducted using excel.

Line 460: there should be additional supplementary tables with the raw data of all measures/data reported on.

6. PLOS authors have the option to publish the peer review history of their article (what does this mean?). If published, this will include your full peer review and any attached files.

Reviewer #1: No

Reviewer #2: No

---

## [Author Response · Author response to Decision Letter 0]

7 Aug 2024

Dear Editor-in-Chief, Reviewer 1 and Reviewer 2!

Thank you for the time and effort spent reviewing our manuscript number PONE-D-24-13353 (The effect of seasonal temperatures on the aging of the overwintered honey bee) and suggesting some important points to consider. 

We corrected the manuscript according to the recommendations in the Submission Guidelines during the final step of our editing process (the Materials chapter was placed after the Introduction, citations in the text were numbered, and the References were updated). As a result, all figure numbers were changed before resubmission. However, we responded to the reviewer's comments using the previous numbering.

Our replies are marked by green color after the symbol R (reply) and question numbers for the reviewer’s comments. Also, we copied these modifications in response for convenience when they were available. 

Please, find in the file "Response to Reviewers" the reviewer comments (black) and our response (green)

We hope you will do the needful by considering the above paper for publication in your esteemed Journal – PLoS ONE.

Sincerely,

Professor Hyung Wook Kwon

College of Life Science and Bioengineering, Division of Life Science

Director of the Convergence Research Center for Insect Vectors

President of the Korean Society for Pestology and Disinfection (KSPD)

Incheon National University, 119 Academy-ro, Yeonsu-gu, Incheon, 22012, Republic of Korea

 +82-32-835-0764 (fax); +82-10-3379-6727 (mobile); +82-32-835-8090 (work); 

 Email: hwkwon@inu.ac.kr

---

## [Decision Letter · Decision Letter 1]

29 Aug 2024

PONE-D-24-13353R1The effect of seasonal temperatures on the aging of the overwintered honey beePLOS ONE

Dear Dr. Kwon,

Thank you for submitting your revised manuscript to PLOS ONE. Both reviewers agree that it is improved but still have several major concerns and suggestions for improvement. Therefore, I would like to send it back to you to carefully go through the remaining issues.

We look forward to receiving your revised manuscript.

Kind regards,

Olav Rueppell

Academic Editor

PLOS ONE

Reviewers' comments:

Reviewer's Responses to Questions

**Comments to the Author**

1. If the authors have adequately addressed your comments raised in a previous round of review and you feel that this manuscript is now acceptable for publication, you may indicate that here to bypass the “Comments to the Author” section, enter your conflict of interest statement in the “Confidential to Editor” section, and submit your "Accept" recommendation.

Reviewer #1: (No Response)

Reviewer #2: (No Response)

2. Is the manuscript technically sound, and do the data support the conclusions?

Reviewer #1: Yes

Reviewer #2: Partly

3. Has the statistical analysis been performed appropriately and rigorously? 

Reviewer #1: No

Reviewer #2: Yes

4. Have the authors made all data underlying the findings in their manuscript fully available?

Reviewer #1: Yes

Reviewer #2: Yes

5. Is the manuscript presented in an intelligible fashion and written in standard English?

Reviewer #1: No

Reviewer #2: Yes

6. Review Comments to the Author

Reviewer #1: This version of the manuscript is greatly improved. I thank the authors for their extensive effort to revise it.

Having said that, there remain substantial issues that should be addressed.

1. There remain many wordings that are not clear, and minor technical issues. Some of these I commented on in the first draft but the same issues remain! For example, I commented on the overuse of "honey bee" when "bee" or "it" would be simpler. The new version is better, but the term "honey bee" remains used far too much.

Another example: I am confused by sample sizes in the paragraph in the Methods related to HP gland measurement. It is stated that 90 acini were measured and 10 acini/bee, which equals only 9 bees studied. But it also states that there were 30 bees/colony X 9 colonies = 270 bees! Please have an independent person read the manuscript for clarity.

As before, I have marked many issues and have suggested changes that I think will improve the manuscript. I hope the authors will take them to heart, as they surely want their manuscript to be as strong as possible.

2. In the first draft, I took issue with the HP glands of winter bees being "hypertrophied". I was absolutely wrong on this, and appreciate the authors defending their wording.

3. In the original version, the authors had used "acinus" (singular) when in fact they meant to use "acini" (plural). It seems that in making corrections, they changed a correct term, "acinus size", to the incorrect term "acini area". Here, "acinus" (singular) is correct (just as one would write "We measured bee size", not "We measured bees size"). And I prefer "acinus size" rather than "acinus area", but that is up to the authors to decide on.

4. Statistics: There is at least one instance where I believe average values were analyzed rather than the raw data. For example, when measuring acinus size, (line 185, revised doc), if I understand the methodology, the authors averaged the areas for the 10 bees from a colony, then conducted the analysis on the average values. Doing this will eliminate much of the variation upon which the analysis of variance is based.

Reviewer #2: The authors indeed improved the manuscript in comparison to the previous version. They incorporated many aspects as suggested. Many aspects and parts of the manuscript still need improvements and corrections.

Major comment

I suggest the authors again take their results and think deeply about their meaning in respect to temperature, nutrition and season. I advise to first stick to temperature, nutrition and season (the factors accounted for in the experiment) and only later in the discussion start to give a putative link to age.

The authors should take a step back and consider how the results of the different physiological and molecular markers can be linked and interpreted, which picture does emerge. At the moment I find the results and especially the discussion highly confusing since the focus constantly shifts between treatment, age, nutrition, season,... their connection should become clearer.

The clear link between the results found for the different factors is missing. E.g. do they hint in the same direction or not? I know that this is not easy because several factors are putatively associated with multiple effects, but what is the take home message? What are the results from this study and what kind of interpretation is possible?

Minor

Abstract

Line 35: what do you mean by "expected markers"?

Line 37: in your study, you simulate "climate warming". I wonder whether it wouldn't make more sense to report that increasing the temperature to 35C leads to faster physiological aging. Thus argument vice versa, since this is what you are actually interested in and what you want to show with this study, right?

Line 39: add "physiological and molecular markers" instead of just markers

Line 40: change to "with an upregulation"

Introduction

Line 61: this should be "foraging"

Line 60-61: are these studies specifically referring to climate warming? Expect for rodents these are all ectotherms and in all ectotherms lifespan depends on temperature...

Line 62: how do "beekeeper management practices [12, 13], and pesticide exposure from land use" link to climate change? These are factors that affect bees irrespective of climate change? Or do you want to say that the impact of these factors might become even stronger under climate change?

Line 64: "mitigate the impacts of climate change and prevent honey bee winter mortality are currently lacking" this sounds like you are testing methods to prevent winter mortality? Please rephrase this sentence and don't rise wrong expectations. This may be your final goal, but it goes beyond this study.

Line 95: delete "and impacts internal processes"

Line 104: change "which" to "with"

Line 123: wouldn't "lipid content" be more appropriate?

Line 152: "testing them together requires further investigation" what exactly do you mean here? Testing what together? And why further investigations? Do you mean that testing the combined effect of the genes is the aim of this study?

Line 162: delete "the last"

Line 168: How about writing "the aging status by means of physiological and molecular markers"?

After reading the manuscript again I would suggest to write "and to compare the effect of summer temperature on the physiological and molecular state of winter bees".

Line 176/177: this should be "early aging" and I find the conclusion here highly far fetched. I propose to delete this part.

You did not explicitely test whether the physiological and molecular you observe indeed result in differences in aging. I still miss the mentioning of the physiological cost of brood feeding which also may lead to a reduction in lifespan. This has not been taken into account in this study, e.g. brood area to number of bees.

To what extent would this lead to changes in the traits measured here? => all this would be necessary to take into account for a "testkit".

Let's assume I am a beekeeper and use the kit in February to measure the aging status of my bees. What measures can I take in February if I find out that my bees are in an advanced aging status?

Line 183: change "housed" to "contained"

Line 184: change "and beginning egg-laying" to "which started to lay eggs in "

Line 186: number of bees missing

Line 187: rephrase "populations"

Line 200: add "per colony"

Line 203: change "Throughout the previous summer" to "Througout summer"

Line 204: change "varying room" to "the according"

Line 205/206: Rephrase. E.g. "From ... onwards, all colonies were kept under natural conditions.

Line 220: do you mean three measurements "per" minute? = 3x60x24= 4320 measuers/day? What do you mean by approximately one file per day? Is the file information relevant? Maybe just delete.

So it turns out that the microclimate was only monitored for 2 weeks in summer. This is super relevant information and should be stated in sections where these numbers are used.

Why was the microclimate only recorded in summer over 15days and not throughout the experiement?

Having this information from two weeks in summer only, does not allow us to draw sound conclusions about transcriptomes obtained from bees in February. Many temperature changes could have happened in the meantime... maybe the insulation of the three rooms differs,...?

Line 247: change "bee" to plural "bees"; delete "Apis mellifera ligustica colonies"

Line 256: "Total tissue RNA (brain or abdomen) was extracted from fifteen randomly selected" I don't understand this procedure. Why 15 random bees, and what do you mean by random? Could this result in 10 bees from a single colony and five bees from a second?

You took 10 bees from each of nine colonies, why didn't you take one bee per colony for each of the measures? Or one bee per colony for RNA tissue extraction of the brain and fat body and another bee for the lipid content?

Line 258: this is redundant to line 254, delete.

Line 311: "bounded" I guess this should be "bound"

Line 313++: in my opinion it is not correct to perform this test here based on the data obtained from the experiment. To conduct such a test you would first have to perform an experiment which would result in data suitable for this test. I.e. you would have to perform an experiment in which you explicitly investigate indviduals with different acini sizes and lipid content, which are otherwise kept under the same conditions. Here in this experiement you had 3 different temperature regimes and the question is whether these different treatments affect physiological and/or molecular markers. But to my point of view your data does not allow to test for an effect of gene expression on acinius size of lipid content. These could all be correlations due to your experimental setup.

I suggest to remove the ENR from M&M plus the results.

Results

Line 353: "Although significant temperature differences were not 353 detected, a stable elevated night temperature is one of the factors of climate warming 354 that influences bees." I don't understand what the authors want to convey here? Please clarify.

Lines 355 and 358: change "same tests" to "ANOVA".

Plus I still have the same comment and questions as in the previous round of review concerning the <0.05 values. And this accounts for all "p < 0.05" throughout the results. It is not sufficient to state p < 0.05 if your p is smaller than than alpha 0.05. You need to be more precise!! I don't believe that all p-values you obtained range between 0.01-0.05 which would be the margin represented by 0.05. You must have some values which are smaller than that which should either be indicated by the exact p-value or by 0.01 (if between 0.001-0.01) or p<0.001.

Line 362: here you start writing bees instead of "honey bees" but up to here always "honey bees". Writing bees is fine, just use it consistent throughout the manuscript, i.e. change above.

Line 358: you should reference to Fig3 here.

Line 361: I cannot find information in the M&M how you measured number of larvae and capped brood area? How was this measured, how many combs, which area? What do you mean by honey bees’ measurements?

Line 366: "Groups control and 25 on August 08 (Fig 3 D, E, F) showed no significant differences,", why was August 8th chosen as the reference date to make the decision on putting loggers into hives or not.

Yes the referred numbers did not differ in August, but when we look at the last September measure the control group differs in all three measures from the 25 and 35 group, which seem to have significantly lower number of larvae, capped brood cells and bees.

Figure 3: A and B remove "," and put "(°C) and (%)

Line 389: throughout M&M you refer to the candidate genes as genes associated with "six genes related to nutrition (vg, ilp1, ilp2, TOR1),

development (JHAMT), and stress response (HSP70)" it comes out of the blue that these allow to investigate the "biological age". Please make sure that you introduce this link latest in the M&M or stick to the information given in M&M here in the results section.

Line 392: remove "mass of"

Fig4 caption: change "mass" to "lipid content"

Fig5 and according text: as mentioned above, I think the results obtained from this experiment/study do not allow to perform this analysis and the results should be removed. Wouldn't it be possible to perform this analysis with "treatment" as dependent variable to obtain information on the effect of temperature on each of the factors and their importance in desciminating between treatments?

Line 434: I suggest to find a new sub-header. First please make sure that the position and content of subheaders is logical. Currently the previous header fits for all results. You could specify in subheaders that you once look at physiological and then at the molecular markers to discriminate between treatment groups.

Second, I don't think that your experiments really allow to draw sound conclusions about the biological age of bees based on those markers. Feel free discuss this possiblily in the discussion, but your experimental data "only" allows to draw conclusions on the effect different temperature regimes on physiolocial traits and gene expression of candidate genes (which can be associated with...). This by itself is interesting!

To draw sound conclusions on the biological age of bees you would need to experiementally disentangle age from temperature by using different age cohorts, which you do not have. I.e. please be careful and rather remove this statement. Lipid content for example could be related to nutrition only, not age. Maybe bees at higher temperatures have a higher metabolism and thus use up their fat reserves quicker,...???

Line 440: you did not measure fat body mass, you measured lipid content. Be specific here and in the following.

Line 448: don't write "same test", be specific. Also in the following sections, i.e. throughout the manuscript.

Line 452: "accelerated physiologicaly aging" see above comments concerning this interpretation. Change and downword.

Figure 6: fat body mg should be "fat body lipid content (mg)", change other axis labels accordingly also in respect to format, and change figure caption accordingly.

Line 455: see comment to line 434 and other comments in respect to the use of "age".

Line 458: header says age and here you write task-related.... see above comments

Discussion

Line 489: "prolonged climate warming" I am confused. In the results you say that there are no differences between control and group25, and here you say that group 25 and 30 were warming conditions!? If G25 is similar to the control, then the fact that the brood parameters are lower in both G25 and G30 may be a treatment effect of either the insulated rooms or side effect constant temperatures during larval development, or ....

It might also be better to say "It was found that longer periods of higher tempartures/higher, constant tempratures, as expected under climate warming", as climate is measured over decades and centuries.

Line 494: you actually mention tempearture fluctuations here yourself. Please use this information in the previous sentence and argument along those lines.

Line 495++: rephrase. E.g. "While experimental ambient temperatures were set to 10°C difference, the bees effort to achieve optimal developmental temperatures for the bees reduced this difference to about 3°C. Bees in Group 25 heated the brood to approximately 30°C, while the Group 35 bees cooled the hive to 33°C, following well demonstrated known microclimate regulation [21, 22]."

Line 514: "related to the development of physiological and molecular markers in winter bees, which are responsible for the processes of aging and longevity [69]" rephrase.

Line 530: in line with comment to lipid content in the results section, acinus size has also been linked to nutritional state in nurse bees (e.g. doi:10.3791/58261). I.e. it is not exclusively an aging marker. This should be taken into consideration and clearly be mentioned in the discussion for both lipid content and acinus size. Or even better first be discussed in a way that it actually fits the result i.e. temperature treatment had an effect on acinus size but not lipid content and then putative interpretations. Statistical results = pvalues do not need to be repeated in the discussion.

Line 535: why are physiological difference indicatif of diapause conditions, and why only in two out of three? Please explain.

Line 536: what is your explanation for this? => here in the discussion you refer all differences to nutrition but not age. This is not in line with what you write in the results. Also see my comments there.

Line 539: please rephrase this sentence, unclear what "all of this" refers to.

Lines 541-544: These sentences are confusing. How does Vg which is transferred to the HG to be fed to larvae relate suppressed JH and conditions of winter bees? Rephrase and make clearer.

Line 551: vg expression also differs between nurses and foragers, does it differ nutritioinal states? Nurses generally also have a higher fat content, right? Does it also differ in respect to the amount of brood in need of food? These could all be confounding factors, which at least need to be mentioned and discussed in addition to the aging process.

Line 554: see above comments to ENR

Line 564: this is interesting and relevant. Here you could spend some lines to give information on other studies showing this effect and especially in respect to "being already well-known". Well known for what? Be more specific.

Line 577-585: above you write that HSP70 can be stress-associated but it has also be shown to differ between overwintering bees from fall to spring. Here you write about putative stress and nutritional needs. Where do nutritional needs come from? This has not been mentioned before. Please write this section more clearly, and more in line with the other arguments, currently I am lost.

Line 585: delete this sentence.

Line 588: why, how so?

Line 595: could you add some guesses why you do not find an effect if this gene has been associated to nutrition, age and season, all three factors seemingly relevant to this experiment?

Line 600: all of a sudden you write about constant temperatures. I totally agree that this could be a very important factor, but this factor hardly finds any mention and discussion in the discussion. Please add.

Line 606: "In contrast, the 35°C temperature hindered colony development, as honey

606 bees in Group 35 were unable to reduce brood temperatures as effectively as those in

607 Group 25, impairing winter bee development. " this statement should be downworded. Since there are no measures for brood temperature in the control group, it is impossible to say whether the 35 group had suboptimal developmental conditions. Moreover the 25 and 35 group both have less brood and workers in comparison to the control. Thus in my opinion, the results do not support this strong statement and rather hint to suboptimal conditions in the 25 and 30C groups, e.g. because of constant temperatures? Maybe constant high temperatures require the bees to constantly ventilate the hive and thus use up many more resources than the control bees, which in turn leaves them less resources for the brood?

Line 610: where are the numbers for honey bee survival? I did not see a section of measuring honey bee survival in M&M nor any results. this is actually a very good point. The authors marked 40bees per colony, how many of those were still alive in February?

Line 611: I guess you mean summer and not fall? At least you are writing about summer conditions throughout the manuscript... please be consistent. And you should be more specific and write "such as increased summer temperatures" .

Line 613: Why "extreme heat", before it was higher temperatures...? Be consistent.

Unclear how "extreme heat" relates to management in winter? Be more specific.

What kind of management plans do you have in mind? Be more specific.

7. PLOS authors have the option to publish the peer review history of their article (what does this mean?). If published, this will include your full peer review and any attached files.

Reviewer #1: No

Reviewer #2: No

---

## [Author Response · Author response to Decision Letter 1]

4 Oct 2024

Dear Editor-in-Chief, Reviewer 1 and Reviewer 2!

Thank you for the time and effort spent reviewing our manuscript number PONE-D-24-13353R1 (The effect of seasonal temperatures on the physiology of the overwintered honey bee) and suggesting some important points to consider. We corrected the manuscript according to the reviewers' comments. Our replies are marked by blue after the symbol R (reply) and question numbers for the reviewer’s comments. Also, we copied these modifications in response for convenience when they were available. 

Please, find below in attached file "Response to Reviewers" the reviewer comments (black) and our response (blue).

We would like to thank the reviewers for their careful and thorough reading of this manuscript and for the thoughtful comments and constructive suggestions, which helped to improve the quality of this manuscript. 

Sincerely,

Professor Hyung Wook Kwon

Department of Life Science, 

Convergence Research Center for Insect Vectors

Incheon National University, R&D Complex, 

265 Harmony-ro, Yeonsu-gu, Incheon, 22014, Republic of Korea; 

+82-32-835-8090 (work); 

+82-10-3379-6727 (mobile);

+82-32-835-0764 (fax). 

Email: hwkwon@inu.ac.kr

---

## [Decision Letter · Decision Letter 2]

15 Nov 2024

PONE-D-24-13353R2The effect of seasonal temperatures on the physiology of the overwintered honey beePLOS ONE

Dear Dr. Kwon,

Thank you for resubmitting your manuscript to PLOS ONE. The new version is much improved and I only send it out to one reviewer. However, this reviewer identifies still a lot of points that need improvement and I agree. These are mainly language issues, but it is important for the readers to unambiguously understand your writing. Please go carefully through the manuscript once more to improve its readability.

We look forward to receiving your revised manuscript.

Kind regards,

Olav Rueppell

Academic Editor

PLOS ONE

Journal Requirements:

Reviewers' comments:

Reviewer's Responses to Questions

**Comments to the Author**

1. If the authors have adequately addressed your comments raised in a previous round of review and you feel that this manuscript is now acceptable for publication, you may indicate that here to bypass the “Comments to the Author” section, enter your conflict of interest statement in the “Confidential to Editor” section, and submit your "Accept" recommendation.

Reviewer #1: (No Response)

2. Is the manuscript technically sound, and do the data support the conclusions?

Reviewer #1: Yes

3. Has the statistical analysis been performed appropriately and rigorously? 

Reviewer #1: Yes

4. Have the authors made all data underlying the findings in their manuscript fully available?

Reviewer #1: Yes

5. Is the manuscript presented in an intelligible fashion and written in standard English?

Reviewer #1: No

6. Review Comments to the Author

Reviewer #1: Overall the manuscript is clear. I commend the authors for making many changes that have improved its quality.

I indicated "No" to Question 5 to clarify that quite a few editorial changes are still required before this manuscript should be published. Most of my comments and changes are minor, but there quite a few places where the wording could be improved or the grammar is not correct. Additionally, there are a few sentences that I could not understand.

I have made my suggestions and corrections on the manuscript document (attached pdf file). They can be found throughout the manuscript into the references. For the record, after the first round of reviews there should be no errors in the references. The quality of the final manuscript reflects on all of the authors. It is important to have someone, an author or other editor, review your work and check it carefully so the final "product" will be of the highest possible quality.

7. PLOS authors have the option to publish the peer review history of their article (what does this mean?). If published, this will include your full peer review and any attached files.

Reviewer #1: No

---

## [Author Response · Author response to Decision Letter 2]

18 Nov 2024

Dear Editor-in-Chief, Reviewer 1 and Reviewer 2!

Thank you for the time and effort spent reviewing our manuscript number PONE-D-24-13353_R2-1 Nov (The effect of seasonal temperatures on the physiology of the overwintered honey bee) and suggesting very important points to consider. We corrected the manuscript according to the recommendations.

We have addressed all the points and made the necessary corrections in the manuscript. Additionally, we reviewed all the references and made some adjustments. Each correction is highlighted in the 'Reviewed Manuscript with Track Changes.doc.

We are pleased to resubmit for publication the manuscript “The effect of seasonal temperatures on the physiology of the overwintered honey bee”. We would like to thank the reviewers for their careful and thorough reading of this manuscript and for the thoughtful comments and constructive suggestions, which helped to improve the quality of this manuscript. 

Sincerely,

Professor Hyung Wook Kwon 

College of Life Science and Bioengineering, Division of Life Science

Director of the Convergence Research Center for Insect Vectors

President of the Korean Society for Pestology and Disinfection (KSPD)

(new address, because the lab was moved to the new building): Incheon National University, R&D Complex, 265 Harmony-ro, Yeonsu-gu, Incheon, 22014, Republic of Korea 

+82-32-835-0764 (fax); +82-10-3379-6727 (mobile); +82-32-835-8090 (work); 

 Email: hwkwon@inu.ac.kr

---

## [Editor Report · Decision Letter 3]

21 Nov 2024

The effect of seasonal temperatures on the physiology of the overwintered honey bee

PONE-D-24-13353R3

Dear Dr. Kwon,

We’re pleased to inform you that your manuscript has been judged scientifically suitable for publication and will be formally accepted for publication once it meets all outstanding technical requirements.

Kind regards,

Olav Rueppell

Academic Editor

PLOS ONE
---

## [Editor Report · Acceptance letter]

28 Nov 2024

PONE-D-24-13353R3 

PLOS ONE

Dear Dr. Kwon, 

I'm pleased to inform you that your manuscript has been deemed suitable for publication in PLOS ONE. Congratulations! Your manuscript is now being handed over to our production team.

Kind regards, 

on behalf of

Dr. Olav Rueppell 

Academic Editor

PLOS ONE